# DäRF: Boosting Radiance Fields from Sparse Inputs with Monocular Depth Adaptation

**Jiuhn Song**[*]    **Seonghoon Park**[*]    **Honggyu An**[*]
**Seokju Cho**    **Min-Seop Kwak**    **Sungjin Cho**    **Seungryong Kim**[†]

Korea University

## Abstract

Neural radiance field (NeRF) shows powerful performance in novel view synthesis and 3D geometry reconstruction, but it suffers from critical performance degradation when the number of known viewpoints is drastically reduced. Existing works attempt to overcome this problem by employing external priors, but their success is limited to certain types of scenes or datasets. Employing monocular depth estimation (MDE) networks, pretrained on large-scale RGB-D datasets, with powerful generalization capability would be a key to solving this problem: however, using MDE in conjunction with NeRF comes with a new set of challenges due to various ambiguity problems exhibited by monocular depths. In this light, we propose a novel framework, dubbed DäRF, that achieves robust NeRF reconstruction with a handful of real-world images by combining the strengths of NeRF and monocular depth estimation through online complementary training. Our framework imposes the MDE network's powerful geometry prior to NeRF representation at both seen and unseen viewpoints to enhance its robustness and coherence. In addition, we overcome the ambiguity problems of monocular depths through patch-wise scale-shift fitting and geometry distillation, which adapts the MDE network to produce depths aligned accurately with NeRF geometry. Experiments show our framework achieves state-of-the-art results both quantitatively and qualitatively, demonstrating consistent and reliable performance in both indoor and outdoor real-world datasets. Project page is available at `https://ku-cvlab.github.io/DaRF/`.

## 1   Introduction

Neural radiance field (NeRF) [27] has gained significant attention for its powerful performance in reconstructing 3D scenes and synthesizing novel views. However, despite its impressive performance, NeRF often comes with a considerable limitation in that its performance highly relies on the presence of densely well-calibrated input images which are difficult to acquire. As the number of input images is reduced, NeRF's novel view synthesis quality drops significantly, displaying failure cases such as erroneous overfitting to the input images [15, 29], artifacts clouding empty spaces [29], or degenerate geometry that yields incomprehensible jumble when rendered at unseen viewpoints [16]. These challenges derive from its under-constrained nature, causing it to have extreme difficulty mapping a pixel in input images to a correct 3D location. In addition, NeRF's volume rendering allows the model to map a pixel to multiple 3D locations [11], exacerbating this problem.

Previous *few-shot* NeRF methods attempt to solve these issues by imposing geometric regularization [29, 16, 20] or exploiting external 3D priors [11, 37] such as depth information extracted from

---

[*]Equal contribution
[†]Corresponding author

37th Conference on Neural Information Processing Systems (NeurIPS 2023).

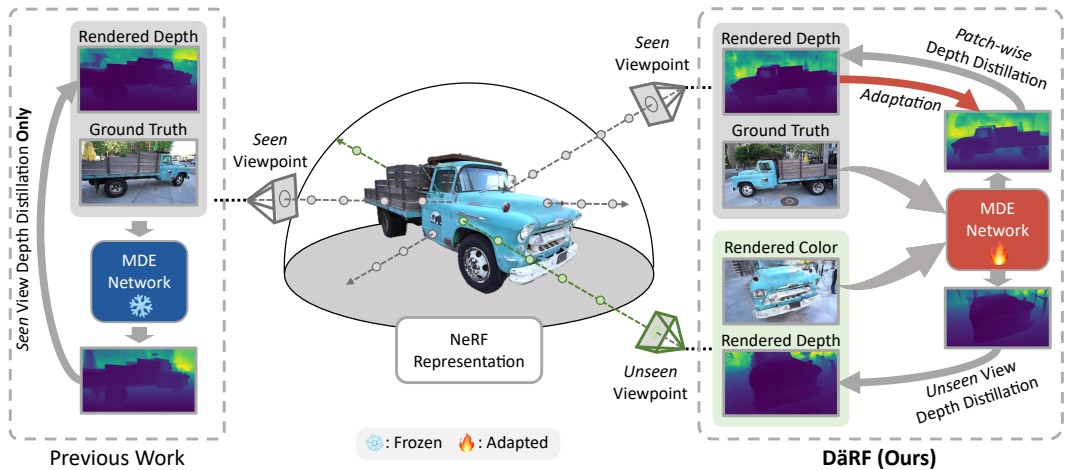

Figure 1: **Overview.** DäRF shows robust optimization of few-shot NeRF through MDE's geometric prior, removing inherent ambiguity from MDE through novel patch-wise distillation loss and MDE adaptation. Unlike existing work [45] that distills depths by applying pretrained MDE to NeRF at seen view only, our DäRF fully exploits the ability of MDE by jointly optimizing NeRF and MDE at a specific scene, and distilling the monocular depth prior to NeRF at both seen and unseen views.

input images by COLMAP [39]. However, these methods have weaknesses in that they use 3D priors extracted from a few input images only, which prevents such guidance from encompassing the entire scene. To effectively tackle all the issues mentioned above, pretrained monocular depth estimation (MDE) networks with strong generalization capability [35, 34, 4] could be used to inject an additional 3D prior into NeRF that facilitates robust geometric reconstruction. Specifically, geometry prediction by MDE can constrain NeRF into recovering smooth and coherent geometry, while their bias towards predicting smooth geometry helps to filter out fine-grained artifacts that clutter the scene. More importantly, NeRF's capability to render any unseen viewpoints enables fully exploiting the capability of the MDE, as MDE could provide depth prior to the numerous renderings of unseen viewpoints as well as the original input viewpoints. This allows injecting additional 3D prior to effectively covering the entire scene instead of being constrained to a few input images.

However, applying MDE to few-shot NeRF is not trivial, as there are ambiguity problems that hinder the monocular depth from serving as a good 3D prior. Primarily, relative depths predicted by MDEs are not multiview-consistent [5]. Moreover, MDEs perform poorly in estimating depth differences between multiple objects: this prevents global scale-shift fitting [55, 26] from being a viable solution, as alignment to one region of the scene inevitably leads to misalignment in many other regions. There also exists a convexity problem [26], in which the MDE has difficulty determining whether the surface is planar, convex, or concave, are also present. To overcome these challenges, we introduce a novel method to adapt MDE to NeRF's absolute scaling and multiview consistency as NeRF is regularized by MDE's powerful 3D priors, creating a complementary cycle.

In this paper, we propose DäRF, short for Monocular **D**epth **A**daptation for boosting **R**adiance **F**ields from Sparse Input Views, which achieves robust optimization of few-shot NeRF through MDE's geometric prior, as well as MDE adaptation for alignment with NeRF through complementary training (see Fig. 1). We exploit MDE for robust geometry reconstruction and artifact removal in both *unseen* and *seen* viewpoints. In addition, we leverage NeRF to adapt MDE toward multiview-consistent geometry prediction and introduce novel patch-wise scale-shift fitting to more accurately map local depths to NeRF geometry. Combined with a confidence modeling technique for verifying accurate depth information, our method achieves state-of-the-art performance in few-shot NeRF optimization. We evaluate and compare our approach on real-world indoor and outdoor scene datasets, establishing new state-of-the-art results for the benchmarks.

## 2 Related Work

**Neural radiance field.** Neural radiance field (NeRF) [27] represents photo-realistic 3D scenes with MLP. Owing to its remarkable performance, there has been a variety of follow-up studies [2, 51, 25].

These studies improve NeRF such as dynamic and deformable scenes [31, 44, 33, 1], real-time rendering [51, 36, 28], unbounded scene [3, 42, 48] and generative modeling [40, 30, 7]. However, these works still encounter challenges in synthesizing novel views with a limited number of images in a single scene, limiting their applicability in real-world scenarios.

**Few-shot NeRF.** Numerous *few-shot* NeRF works attempted to address few-shot 3D reconstruction problem through various techniques, such as pretraining external priors [52, 9], meta-learning [43], regularization [15, 29, 16, 20] or off-the-shelf modules [15, 29]. Recent approaches [29, 16, 20] emphasize the importance of geometric consistency and apply geometric regularization at unknown viewpoints. However, these regularization methods show limitations due to their heavy reliance on geometry information recovered by NeRF. Other works such as DS-NeRF [11], DDP-NeRF [37] and SCADE [45] exploit additional geometric information, such as COLMAP [39] 3D points or monocular depth estimation, for geometry supervision. However, these works have critical limitations of only being able to provide geometry information corresponding to existing input viewpoints. Unlike these works, our work demonstrates methods to provide geometric prior even at unknown viewpoints with MDE for more effective geometry reconstruction.

**Monocular depth estimation.** Monocular depth estimation (MDE) is a task that aims to predict a dense depth map given a single image. Early works on MDE used handcrafted methods such as MRF for depth estimation [38]. After the advent of deep learning, learning-based approaches [14, 17, 21] were introduced to the field. In this direction, the models were trained on ground-truth depth maps acquired by RGB-D cameras or LiDAR sensors to predict absolute depth values [24, 23]. Other approaches trained the networks on large-scale diverse datasets [8, 22, 34, 35], which demonstrates better generalization power. These approaches struggle with depth ambiguity caused by ill-posed problem, so the following works LeRes [50] and ZoeDepth [4] opt to recover absolute depths using additional parameters.

**Incorporating MDE into 3D representation.** As both NeRF and monocular depth estimation are closely related, there have been some works that utilize MDE models to enhance NeRF's performance. NeuralLift [49], MonoSDF [53] and SCADE [45] leverage depths predicted by pretrained MDE for depth ordering and detailed surface reconstruction, respectively. Other works optimize scene-specific parameters, such as depth predictor utilizing depth recovered by COLMAP [47] or learnable scale-shift values for reconstruction in noisy pose setting [6]. However, these previous approaches were limited in that MDEs were used to provide prior to only the input viewpoints, which constrains their effectiveness when input views are reduced, e.g., in the few-shot setting.

As a concurrent work, SCADE [45] utilizes MDE for sparse view inputs, by injecting uncertainty into MDE through additional pretraining so that canonical geometry can be estimated through probabilistic modeling between multiple modes of estimated depths. While the ultimate goal which is to overcome the ambiguity of MDE may be similar, our approach directly removes ambiguity present in MDE by finetuning with canonical geometry captured by NeRF, for effective suppression of artifacts and divergent behaviors of few-shot NeRF.

## 3 Preliminaries

NeRF [27] represents a scene as a continuous function $\mathcal{F}_\theta(\cdot)$ represented by a neural network with parameters $\theta$. During optimization, 3D points are sampled along rays represented by $\mathbf{r}$ coming from a set of input images $\mathcal{S} = \{I_i\}$, whose ground truth camera poses are given, for evaluation by the neural network. For each sampled point, $\mathcal{F}_\theta(\cdot)$ takes as input its coordinate $\mathbf{x} \in \mathbb{R}^3$ and viewing direction $\mathbf{d} \in \mathbb{R}^2$ with a positional encoding $\gamma(\cdot)$ that facilitates learning high-frequency details, and outputs a color $\mathbf{c} \in \mathbb{R}^3$ and a density $\sigma \in \mathbb{R}$ such that $\{\mathbf{c}, \sigma\} = \mathcal{F}_\theta(\gamma(\mathbf{x}), \gamma(\mathbf{d}))$. With a ray parameterized as $\mathbf{r}_\mathbf{p}(t) = \mathbf{o} + t\mathbf{d}_\mathbf{p}$, starting from camera center $\mathbf{o}$ along the direction $\mathbf{d}_\mathbf{p}$, color and depth value at the pixel $\mathbf{p}$ are rendered as follows:

$$\bar{I}(\mathbf{p}) = \int_{t_n}^{t_f} T(t)\sigma(\mathbf{r}_\mathbf{p}(t))\mathbf{c}(\mathbf{r}_\mathbf{p}(t))dt, \quad \bar{D}(\mathbf{p}) = \int_{t_n}^{t_f} T(t)\sigma(\mathbf{r}_\mathbf{p}(t))t dt, \tag{1}$$

where $\bar{I}(\mathbf{p})$ and $\bar{D}(\mathbf{p})$ are rendered color and depth values at the pixel $\mathbf{p}$ along the ray $\mathbf{r}_\mathbf{p}(t)$ from $t_n$ to $t_f$, and $T(t)$ denotes an accumulated transmittance along the ray from $t_n$ to $t$ as follows:

$$T(t) = \exp\left(-\int_{t_n}^{t} \sigma(\mathbf{r}_\mathbf{p}(s))ds\right). \tag{2}$$

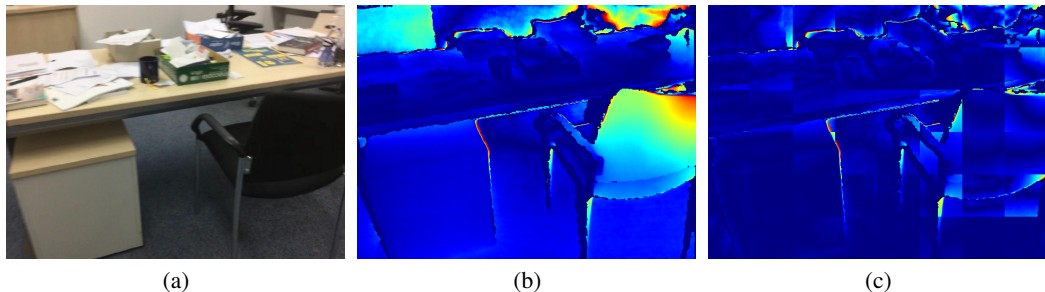

|     (a)     |     (b)     |     (c)     |

Figure 2: **Effectiveness of patch-wise scale and shift adjustment:** (a) input image, (b) monocular depth with image-level adjustment, and (c) monocular depth with patch-level adjustment. We visualize the error of adjusted monocular depth from input image compared to GT depth value. The proposed patch-level adjustment helps to minimize the errors caused by inconsistency in depth differences among objects.

Based on this volume rendering, $\mathcal{F}_\theta(\cdot)$ is optimized by the reconstruction loss $\mathcal{L}_{\text{recon}}$ that compares rendered color $\bar{I}(\mathbf{p})$ to corresponding ground-truth $I(\mathbf{p})$, with $\mathcal{R}$ as a set of pixels for training rays:

$$\mathcal{L}_{\text{recon}} = \sum_{I_i \in \mathcal{S}} \sum_{\mathbf{p} \in \mathcal{R}} \|I_i(\mathbf{p}) - \bar{I}_i(\mathbf{p})\|_2^2. \tag{3}$$

Our work explores the setting of few-shot optimization with NeRF [16, 20]. Whereas the number of input viewpoints $|\mathcal{S}|$ is normally higher than one hundred in the standard NeRF setting [27], the task of few-shot NeRF considers scenarios when $|\mathcal{S}|$ is drastically reduced to a few viewpoints (e.g., $|\mathcal{S}| < 20$). With such a small number of input viewpoints, NeRF shows high divergent behaviors such as geometry breakdown, overfitting to input viewpoints, and generation of artifacts that cloud the empty space between the camera and object, which causes its performance to drop sharply [15, 16, 29]. To overcome this problem, existing few-shot NeRF frameworks applied regularization techniques at unknown viewpoints to constrain NeRF with additional 3D priors [37, 11] and enhance the robustness of geometry, but they showed limited performance.

## 4    Methodology

### 4.1    Motivation and Overview

Our framework leverages the complementary benefits of few-shot NeRF and monocular depth estimation networks for the goal of robust 3D reconstruction. The benefits that pretrained MDE can provide to few-shot NeRF are clear and straightforward: because they predict dense geometry, they provide guidance for the NeRF to recover more smooth geometry. In cases where few-shot NeRF's geometry undergoes divergent behaviors, MDE provides strong constraints to prevent the global geometry from breaking down.

However, there are difficult challenges that must be overcome if the depths estimated by MDE are to be used as 3D prior to NeRF. These challenges, which can be summarized as depth ambiguity problems [26], stem from the inherent ill-posed nature of the monocular depth estimation. Most importantly, MDE networks only predict relative depth information inferred from an image, meaning it is initially not aligned to NeRF's absolute geometry [4]. Global scaling and shifting may seem to be the answer, but this approach leads us to another depth ambiguity problem, as predicted scales and spacings of each instance are inconsistent with one another, as demonstrated in (b) of Fig. 2. Additionally, MDE's weakness in predicting the convexity of a surface, whether it is flat, convex, or concave - also poses a problem in using this depth for NeRF guidance.

In this light, we adapt a pre-trained monodepth network to a single NeRF scene so that its powerful 3D prior can be leveraged to its maximum capability in regularizing the few-shot NeRF. In the following, we first explain how to distill geometric prior from off-the-shelf MDE model [35] from both seen and unseen viewpoints (Sec. 4.2). We also provide a strategy for adapting the MDE model to handle ill-posed problems to a specific scene, while keeping its 3D prior knowledge (Sec. 4.3). Then, we demonstrate a method to handle inaccurate depths (Sec. 4.4). Fig. 1 shows an overview of our method, compared to previous works using MDE prior [45, 53].

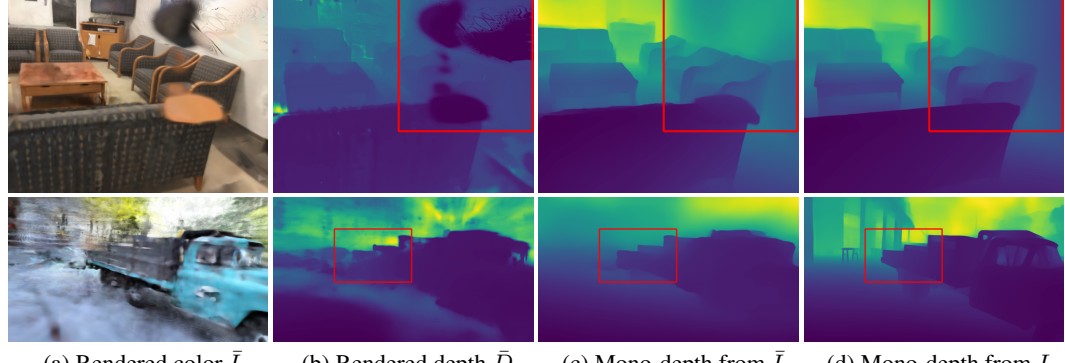

| (a) Rendered color $\bar{I}$ | (b) Rendered depth $\bar{D}$ | (c) Mono-depth from $\bar{I}$ | (d) Mono-depth from $I$ |

Figure 3: **Robustness of MDE model for multi-view scale ambiguity and artifacts:** (a-b) color and depth of NeRF rendered in the early stage of the training, (c-d) monocular depths estimated from rendered image $\bar{I}$ and input image $I$. The results show that MDE model ignores the artifacts of rendered images by NeRF, enabling reliable supervision for seen and unseen viewpoint.

## 4.2 Distilling Monocular Depth Prior into Neural Radiance Field

To prevent the degradation of reconstruction quality in few-shot NeRF, we propose to distill monocular depth prior to the neural radiance field during optimization. By exploiting pre-trained MDE networks [34, 35], which have high generalization power, we enforce a dense geometric constraint on both *seen* and *unseen* viewpoints by using estimated monocular depth maps as pseudo ground truth depth for training few-shot NeRF. We describe the details of this process below.

**Monocular depth regularization on seen views.** We leverage a pre-trained MDE model, denoted as $\mathcal{G}_\phi(\cdot)$ with parameters $\phi$, to predict pseudo depth map from given *seen* view image $I_i$ as $D_i^* = \mathcal{G}_\phi(I_i)$. Since $D_i^*$ is initially a relative depth map, it needs to be scaled and shifted into an absolute depth [55] and aligned with NeRF's rendered depth $\bar{D}$ in order for it to be used as pseudo-depth $D^*$. However, the scale and shift parameters inferred from the global statistic may undermine local statistic [55]. For example, as shown in Fig. 2 (b), global scale fitting tends to favor dominant objects in the image, leading to ill-fitted depths in less dominant sections of the scene due to inconsistencies in predicted depth differences between the objects. Naïvely employing such inaccurately estimated depths for distillation can adversely impact the overall geometry of the NeRF.

To alleviate this issue, we propose a patch-wise adjustment of scale and shift parameters, reducing the impact of erroneous depth differences, as illustrated in Fig. 2 (c). The depth consistency loss is defined as follows:

$$\mathcal{L}_{\text{seen}} = \sum_{I_i \in \mathcal{S}} \sum_{\mathbf{p} \in \mathcal{P}} \|(w_i \mathbf{sg}\,(D_i^*(\mathbf{p})) + q_i) - \bar{D}_i(\mathbf{p})\|, \tag{4}$$

where $w_i$ and $q_i$ denote the scale and shift parameters obtained by least square [35] between $D_i^*$ and $\bar{D}_i$, $\mathcal{P}$ denotes a set of pixels within a patch, and $\mathbf{sg}(\cdot)$ denotes stop-gradient operation. Thus patch-based approach also helps to overcome the computational bottleneck of full image rendering.

**Monocular depth regularization on unseen views.** We further propose to give supervision even at *unseen* viewpoints. As NeRF has the ability to render any unseen viewpoint of the scene, we render color $\bar{I}_l$ and depth $\bar{D}_l$ from a sampled patch of $l$-th novel viewpoint. Sequentially, we extract a monocular depth map from the rendered image as $\bar{D}_l^* = \mathcal{G}_\phi(\bar{I}_l)$. Then, we enforce consistency between our rendered depth $\bar{D}_l$ and the monocular depth $\bar{D}_l^*$ of $l$-th novel viewpoint as follows:

$$\mathcal{L}_{\text{unseen}} = \sum_{I_l \in \mathcal{U}} \sum_{\mathbf{p} \in \mathcal{P}} \|(w_l \mathbf{sg}\,(\bar{D}_l^*(\mathbf{p})) + q_l) - \bar{D}_l(\mathbf{p})\|, \tag{5}$$

where $\mathcal{U}$ denotes a set of unseen view images, $w_l$ and $q_l$ denotes the scale and shift parameters used to align $\bar{D}_l^*$ towards $\bar{D}_l$, and $\mathcal{P}$ denotes randomly sampled patch.

A valid concern regarding this approach is that monocular depth obtained from noisy NeRF rendering may be affected by fine-grained rendering artifacts that frequently appear in unseen viewpoints of few-shot NeRF, resulting in noisy and erroneous pseudo-depths. However, we demonstrate in

Fig. 3 that a strong geometric prior within the MDE model exhibits robustness against such artifacts, effectively filtering out the artifacts and thereby providing reliable supervision for the unseen views.

It should be noted that our strategy differs from previous methods [11, 37, 53, 45] that exploit monocular depth estimation [34] and external depth priors such as COLMAP [39]. These methods only impose depth priors upon the input viewpoints, and thus their priors only influence the scene partially due to self-occlusions and sparsity of known views. Our method, on the other hand, enables external depth priors to be applied to any arbitrary viewpoint and thus allows guidance signals to thoroughly reach every location of the scene, leading to more robust and coherent NeRF optimization.

## 4.3 Adaptation of MDE via Neural Radiance Field

Although the patch-wise distillation of monocular depth provides invariance to depth difference inconsistency in MDE, the ill-posed nature of monocular depth estimation often introduces additional ambiguities, such as the inability to distinguish whether the surface is concavity, convexity, or planar or difficulty in determining the orientation of flat surfaces [26]. We argue that these ambiguities arise due to the MDE lacking awareness of the scene-specific absolute depth priors and multiview consistency. To address this issue, we propose providing the scene priors optimized NeRF to MDE, whose knowledge of canonical space and absolute geometry helps eliminate the ambiguities present within MDE. Therefore, we propose to adapt the MDE to the absolute scene geometry, formally written as:

$$\mathcal{L}_{\text{MDE}} = \sum_{I_i \in \mathcal{S}} \sum_{\mathbf{p} \in \mathcal{P}} \left\{ \left\| \text{sg}\left(\bar{D}_i(\mathbf{p})\right) - D_i^*(\mathbf{p}) \right\| + \left\| (w_i \text{sg}\left(\bar{D}_i(\mathbf{p})\right) + q_i) - D_i^*(\mathbf{p}) \right\| \right\}. \tag{6}$$

In addition to the patch-wise loss in Eq. 4, we add an $l$-1 loss without scale-shift adjustment to adapt the MDE with absolute depth prior. We also introduce a regularization term to preserve the local smoothness of MDE, given by:

$$\mathcal{L}_{\text{reg}} = \sum_{I_i \in \mathcal{S}} \sum_{\mathbf{p} \in \mathcal{P}} \left\| (w_i \text{sg}\left(D_i^{*,\text{init}}(\mathbf{p})\right) + q_i) - D_i^*(\mathbf{p}) \right\|, \tag{7}$$

where $D_i^{*,\text{init}}$ denotes monocular depth map of $I_i$ extracted from MDE with initial pre-trained weight.

## 4.4 Confidence Modeling

Our framework must take into account the errors present in both few-shot NeRF and estimated monocular depths, which will propagate [41] and intensify during the distillation process if left unchecked. To prevent this, we adopt confidence modeling [20, 41] inspired by self-training approaches [41], to verify the accuracy and reliability of each ray before the distillation process.

The homogeneous coordinates of a pixel $\mathbf{p}$ in the seen viewpoint are transformed to $\mathbf{p}'$ at the target viewpoint using the viewpoint difference $R_{i \to l}$ and the camera intrinsic parameter $K$, as follows:

$$\mathbf{p}' \sim K R_{i \to l} D_i(\mathbf{p}) K^{-1} \mathbf{p}. \tag{8}$$

We generate the confidence map $M_i$ by measuring the distance between rendered depth of the unseen viewpoint and MDE output of seen viewpoint such that

$$M_i(\mathbf{p}) = \left[ \left\| (w_i D_i^*(\mathbf{p}) + q_i) - \bar{D}_l(\mathbf{p}') \right\| < \tau \right], \tag{9}$$

where $\tau$ denotes threshold parameter, $[\cdot]$ is Iverson bracket, and $D_l(\mathbf{p}')$ refers to depth value of the corresponding pixel at $l$-th unseen viewpoint for reprojected target pixel $\mathbf{p}$ of $i$-th seen viewpoint. We fit $D_i^*$ to absolute scale, where scale and shift parameters, $w_i$ and $q_i$, are obtained by least square [35] between $D_i^*$ and $\bar{D}_i$.

## 4.5 Overall Training

With the incorporation of confidence modeling, the loss functions for both the radiance field and MDE can redefined. $\mathcal{L}_{\text{seen}}$ and $\mathcal{L}_{\text{unseen}}$ can be redefined as:

$$\mathcal{L}_{\text{seen}} = \sum_{I_i \in \mathcal{S}} \sum_{\mathbf{p} \in \mathcal{P}} M_i(\mathbf{p}) \left\| (w_i \text{sg}\left(D_i^*(\mathbf{p})\right) + q_i) - \bar{D}_i(\mathbf{p}) \right\|, \tag{10}$$

$$\mathcal{L}_{\text{unseen}} = \sum_{I_l \in \mathcal{U}} \sum_{\mathbf{p} \in \mathcal{P}} M_l(\mathbf{p}) \left\| (w_l \text{sg}\left(\bar{D}_l^*(\mathbf{p})\right) + q_l) - \bar{D}_l(\mathbf{p}) \right\|. \tag{11}$$

Table 1: **Quantitative comparison on ScanNet [10] and Tanks and Temples [19].** The best results are highlighted in bold, while the second best results are underlined.

| Methods | Depth prior | ScanNet [10] | | | | | | Tanks and Temples [19] | | |
| | | 9 - 10 views | | | 18 - 20 views | | | 10 views | | |
| | | PSNR ↑ | SSIM ↑ | LPIPS ↓ | PSNR ↑ | SSIM ↑ | LPIPS ↓ | PSNR ↑ | SSIM ↑ | LPIPS ↓ |
|---|---|---|---|---|---|---|---|---|---|---|
| NerfingMVS [47] | ✓ | N/A | N/A | N/A | 16.29 | 0.626 | 0.502 | N/A | N/A | N/A |
| $K$-planes [13] | ✗ | 16.01 | 0.618 | 0.494 | 18.70 | 0.708 | 0.400 | 12.57 | 0.453 | 0.607 |
| RegNeRF [29] | ✗ | 16.38 | 0.624 | 0.493 | 18.93 | 0.676 | 0.450 | 14.12 | 0.469 | **0.580** |
| DS-NeRF [11] | ✓ | N/A | N/A | N/A | 20.85 | 0.713 | 0.344 | N/A | N/A | N/A |
| DDP-NeRF [37] | ✓ | N/A | N/A | N/A | 19.29 | 0.695 | 0.368 | N/A | N/A | N/A |
| SCADE [45] | ✓ | **18.83** | 0.646 | **0.375** | 21.54 | 0.732 | **0.292** | 13.46 | 0.402 | 0.607 |
| DäRF (Ours) | ✓ | 18.29 | **0.690** | 0.412 | **21.58** | **0.765** | 0.325 | **15.70** | **0.514** | 0.583 |

Table 2: **Evaluation of depth quality:** (a) quantitative evaluation of the adapted MDE, compared with other monocular depth estimation models and (b) visualization of depth distributions. The adapted MDE by our method shows a similar distribution to that of the ground truth.

| Methods | AbsRel ↓ | SqRel ↓ | RMSE ↓ | RMSE log ↓ |
|---|---|---|---|---|
| LeRes [50] | 0.391 | 0.472 | 0.999 | 0.661 |
| MiDaS [35] | 0.152 | 0.095 | 0.452 | 0.183 |
| DPT [34] | 0.191 | 0.135 | 0.563 | 0.220 |
| DäRF (9 - 10 views) | 0.154 | 0.074 | 0.361 | 0.171 |
| DäRF (18 - 20 views) | **0.151** | **0.071** | **0.356** | **0.168** |

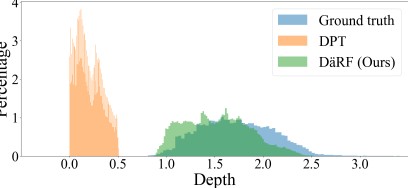

(a) Quantitative comparison  (b) Depth distribution comparison

In addition, the loss for the adaptation of the MDE module can be redefined considering $M$:

$$\mathcal{L}_{\text{MDE}} = \sum_{I_i \in \mathcal{S}} \sum_{\mathbf{p} \in \mathcal{P}} M_i(\mathbf{p}) \left( \| \text{sg} \left( \bar{D}_i(\mathbf{p}) \right) - \bar{D}_i^*(\mathbf{p}) \| + \| (w_i \text{sg} \left( \bar{D}_i(\mathbf{p}) \right) + q_i) - \bar{D}_i^*(\mathbf{p}) \| \right). \quad (12)$$

With these losses, we train both NeRF and MDE simultaneously, enhancing both models by complementing each other. MDE provides a strong geometric prior to NeRF while having the inherent limitation of obliviousness to the scene-specific prior, whereas NeRF provides it with its absolute geometry.

## 5 Experiments

### 5.1 Experimental Settings

**Implementation details.** DäRF is implemented based on $K$-planes [32] as NeRF. We use DPT-hybrid [34] as MDE model. We use Adam [18] as an optimizer, with a learning rate of $1 \cdot 10^{-2}$ for NeRF and $1 \cdot 10^{-5}$ for the MDE, along with a cosine warmup learning rate scheduling. See supplementary material for more details. The code and pre-trained weights will be made publicly available.

**Datasets.** We evaluate our method in real-world scenes captured at both indoor and outdoor locations. Following previous works [37, 45], we use a subset of sparse-view ScanNet data [10] comprised with three indoor scenes, each consisting of 18 to 20 training images and 8 test images. We also conduct evaluations on more challenging setting with 9 to 10 train images. For outdoor reconstruction, we further test on 5 challenging scenes from the Tanks and Temples dataset [19]. The scenes are real-world outdoor dataset, with a wide variety of scene scales and lighting conditions. Note that these setups are extremely sparse compared to full image setups, where we use approximately 0.5 to 5 percent of the whole training inputs.

**Baselines.** We adopt the following six recently proposed methods as baselines: standard neural radiance field method: $K$-planes [13], few-shot NeRF method: RegNeRF [29], and depth prior based methods: NerfingMVS [47], DS-NeRF [11], DDP-NeRF [37], and SCADE [45].

**Evaluation metrics.** For quantitative comparison, we follow the NeRF [27] and report the PSNR, SSIM [46], LPIPS [56]. We report standard evaluation metrics for depth estimation [12], absolute relative error (Abs Rel), squared relative error (SqRel), root mean squared error (RMSE), root mean

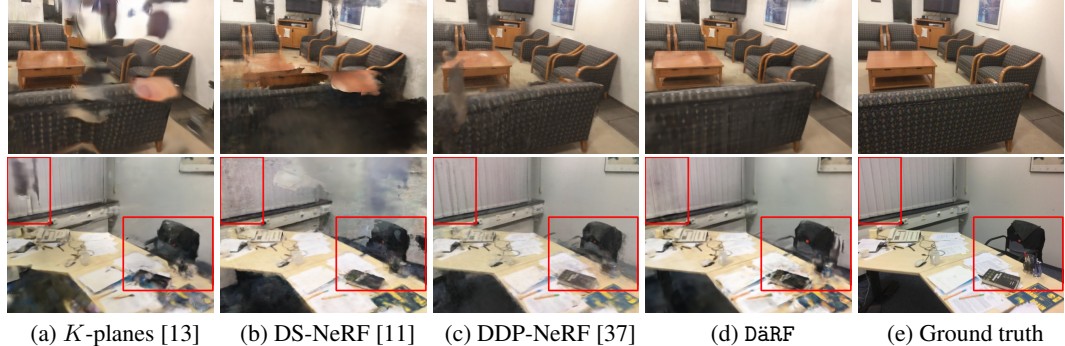

| (a) $K$-planes [13] | (b) DS-NeRF [11] | (c) DDP-NeRF [37] | (d) DäRF | (e) Ground truth |

Figure 5: **Qualitative results of on ScanNet [10] with 18 - 20 input views**.

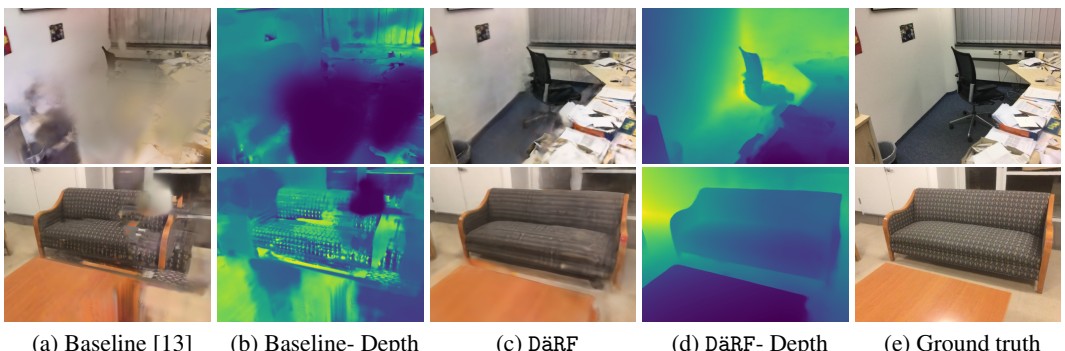

| (a) Baseline [13] | (b) Baseline- Depth | (c) DäRF | (d) DäRF- Depth | (e) Ground truth |

Figure 6: **Qualitative results on ScanNet [10] with 9 - 10 input views**.

squared log error (RMSE log). To evaluate view consistency, we utilize a single scaling factor $s$ for each scene, which is the median scaling [57] value averaged across all test views.

## 5.2 Comparisons

**Indoor scene reconstruction.** We conducted experiments in two settings: (1) a standard few-shot setup as described in literature [37, 45], and (2) an extreme few-shot setup with approximately 0.5 percent of the full images. As shown in Tab. 1, our approach outperforms the baseline methods in both settings in most of the metrics. Additionally, we provide quantitative results of the adapted MDE model in ScanNet dataset in Tab. 2, and qualitative results in Fig. 4. As shown in Fig. 5 for the setting of standard few-shot, DS-NeRF [11] and DDP-NeRF [37] still show floating artifacts in the novel view and show limitation in capturing details in the chair, smoothing into nearby object. Our method shows better qualitative results compared to other baselines, showing better geometry understanding and detailed view synthesis in the small objects near the chair. In the extreme few-shot setup, we conducted

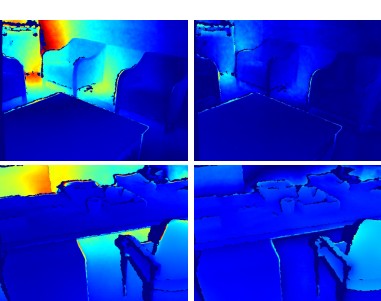

(a) Frozen MDE     (b) Adapted MDE

Figure 4: **Error map visualization.** MDE adaptation results in a reduction of errors.

a visual comparison between our method and a baseline [13] in Fig. 6. This is a more complex setting than standard, but our method outperforms most of the baselines, showing better geometric understanding. It should be noted that SCADE [45] fine-tunes its depth network on an indoor dataset, Taskonomy dataset [54]; whereas our model does not undergo any additional fine-tuning. More qualitative images are included in the supplementary material.

**Outdoor scene reconstruction.** We conduct the qualitative and quantitative comparisons on the Tanks and Temples dataset in Tab. 1 and Fig. 7. Since COLMAP [39] with sparse images is not available, we provide comparisons with baselines without explicit depth prior. The quantitative results show that our approach outperforms the baseline methods on this complex outdoor dataset in all

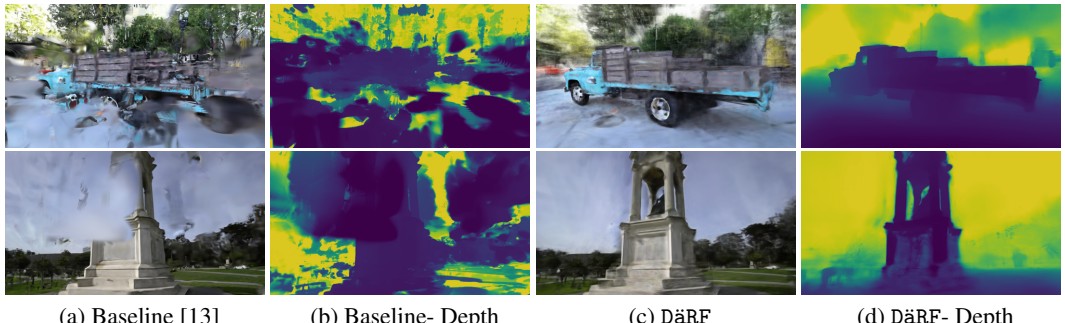

|  |  |  |  |
|:---:|:---:|:---:|:---:|
| (a) Baseline [13] | (b) Baseline- Depth | (c) DäRF | (d) DäRF- Depth |

Figure 7: **Qualitative results on Tanks and Temples [19].**

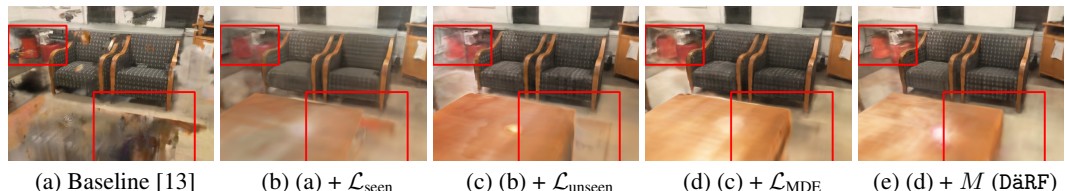

|  |  |  |  |  |
|:---:|:---:|:---:|:---:|:---:|
| (a) Baseline [13] | (b) (a) + $\mathcal{L}_{\text{seen}}$ | (c) (b) + $\mathcal{L}_{\text{unseen}}$ | (d) (c) + $\mathcal{L}_{\text{MDE}}$ | (e) (d) + $M$ (DäRF) |

Figure 8: **Visualization of ablation studies on ScanNet [10].**

metrics. As shown in Fig. 7, our baseline shows limited performance, despite its feasible results of view synthesis in novel viewpoint, its depth results show that the network totally fails to understand 3D geometry. Our method shows rich 3D understanding, even in this real-world outdoor setting which is more complicated than other scenes. More qualitative images are included in the supplementary material.

**In-the-wild dataset Reconstruction.** We further investigate the robustness of our method on in-the-wild dataset, provided from SCADE [45]. This dataset was captured using an iPhoneX, with varying intrinsic parameters for each view. As shown in Tab. 3, our model outperforms all other baselines and this demonstrates that our model performs well in large scenes, such as outdoor scenes and in-the-wild dataset.

Table 3: **Comparison on In-the-wild dataset [45].**

| Methods | PSNR↑ | SSIM↑ | LPIPS↓ |
|---|---|---|---|
| $K$-planes [13] | 20.71 | 0.732 | 0.408 |
| DDP-NeRF [37] | 21.28 | 0.727 | 0.366 |
| SCADE [45] | 22.82 | 0.743 | **0.347** |
| DäRF (Ours) | **22.92** | **0.760** | 0.390 |

## 5.3 Ablation Study

**Ablation on core components.** In Tab. 4 and Fig. 8, we evaluate the effect of each proposed component. The quantitative results show effectiveness of each component. For qualitative results, we found out that $\mathcal{L}_{\text{unseen}}$ suppresses the artifacts in novel viewpoint, compared to when only $\mathcal{L}_{\text{seen}}$ is given. With adaptation of the MDE network to this scene, red basket in the background shows more accurate results and artifacts near the table are removed. In our model, with confidence modeling, view synthesis results show to be more structurally confident in the overall scene.

Table 4: **Ablation study.**

| | Components | PSNR↑ | SSIM↑ | LPIPS↓ |
|---|---|---|---|---|
| **(a)** | Baseline [13] | 18.70 | 0.708 | 0.400 |
| **(b)** | (a) + $\mathcal{L}_{\text{seen}}$ | 19.71 | 0.730 | 0.380 |
| **(c)** | (b) + $\mathcal{L}_{\text{unseen}}$ | 21.21 | 0.758 | 0.333 |
| **(d)** | (c) + MDE Adapt. ($\mathcal{L}_{\text{MDE}}$) | 21.39 | 0.763 | 0.327 |
| **(e)** | (d) + Conf. Modeling (DäRF) | **21.58** | **0.765** | **0.325** |

**Analysis of local fitting.** In Tab. 5, we further investigate the effectiveness of local scale-shift fitting and global scale-fitting. For global scale-shift fitting, we give learnable scale and shift parameters for each input image and convert MDE's output to the absolute value in a global manner. For a fair comparison, we compare our model only with MDE distillation on seen viewpoints. The results show that our method local scale-shift fitting is more effective on giving accurate depth supervision.

Table 5: **Local fitting ablation.**

| Components | PSNR↑ | SSIM↑ | LPIPS↓ |
|---|---|---|---|
| Baseline [13] | 18.65 | 0.706 | 0.502 |
| w/ global fitting | 19.05 | 0.698 | 0.399 |
| w/ local fitting (DäRF) | **19.71** | **0.730** | **0.380** |

Table 6: **Ablation study on MDE Adaptation Loss.**

| Components | PSNR↑ | SSIM↑ | LPIPS↓ | AbsRel↓ | SqRel↓ | RMSE↓ | RMSE log↓ |
|---|---|---|---|---|---|---|---|
| scale-shift loss | 21.31 | 0.757 | 0.343 | 0.182 | 0.109 | 0.484 | 0.205 |
| $l1$ loss | 21.48 | 0.758 | 0.337 | 0.157 | 0.079 | 0.386 | 0.176 |
| DäRF (Ours) | **21.58** | **0.765** | **0.325** | **0.151** | **0.071** | **0.356** | **0.168** |

**Analysis of MDE Adaptation Loss.** In Tab. 6, we further investigate the effectiveness of scale-shift loss and $l1$ loss at MDE adaptation. Equation 6 revolves around the idea of adapting MDE toward predicting a scene-specific absolute geometry, which is achieved by the first addend term: this first term forces itself MDE to adapt towards multiview consistency so that its ill-posed nature is reduced and its initial global depth prediction grows to be more in accordance with the absolute geometry captured by NeRF. In contrast, the second addend term, which takes into account patch-wise scale-shift fitting, is designed to aid the modeling of fine, detailed, local geometry which the model has difficulty modeling without such local fitting. As shown in the results, when only one term is used for optimization (scale-shift or $l1$), it performs worse in every metric than in both are used in conjunction (ours). This justifies our strategy of using both losses as effective.

## 6 Conclusion

We propose DäRF, a novel method that addresses the limitations of NeRF in few-shot settings by fully leveraging the ability of monocular depth estimation networks. By integrating MDE's geometric priors, DäRF achieves robust optimization of few-shot NeRF, improving geometry reconstruction and artifact removal in both unseen and seen viewpoints. We further introduce patch-wise scale-shift fitting for accurate mapping of local depths to 3D space, and adapt MDE to NeRF's absolute scaling and multiview consistency, by distilling NeRF's absolute geometry to monocular depth estimation. Through complementary training, DäRF establishes a strong synergy between MDE and NeRF, leading to a state-of-the-art performance in few-shot NeRF. Extensive evaluations on real-world scene datasets demonstrate the effectiveness of DäRF.

## Acknowledgements

This research was supported by the MSIT, Korea (IITP2022-2020-0-01819, No. 2021-0-02068, No.2021-0-00155), and National Research Foundation of Korea (NRF-2021R1C1C1006897). This research was partially supported by Culture, Sports, and Tourism R&D Program through the Korea Creative Content Agency grant funded by the Ministry of Culture, Sports and Tourism in 2023 (4D Content Generation and Copyright Protection with Artificial Intelligence, R2022020068).

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
