# DäRF: Boosting Radiance Fields from Sparse Inputs with Monocular Depth Adaptation - Supplementary Materials -

## A  Implementation Details

### A.1  Architecture

We implement DäRF with $K$-planes [4] as the base model. It represents a radiance field using tri-planes with three multi-resolutions for each plane: 128, 256, and 512 in both height and width, and 32 in feature depth. This approach also incorporates small MLP decoders and a two-stage proposal sampler. It should be noted that our framework is not restricted to the $K$-planes baseline, but can be incorporated into any NeRF backbone models [9, 10, 8]. In our experiments, we implemented our framework on top of the $K$-planes hybrid version codebase due to its quality, reasonable optimization speed, and model size. For the monocular depth estimation (MDE) module, we choose the pre-trained DPT [12] as our base MDE model due to its powerful generalization ability in a zero-shot setting. Trained on very large datasets, DPT demonstrates impressive prediction quality and generalizes well to novel scenes. However, any MDE model can be utilized within our framework [19, 13, 12].

### A.2  Training details

We use the Adam optimizer [6] and a cosine annealing with warm-up scheduler for NeRF optimization. The learning rate is set to $1 \cdot 10^{-2}$, and we perform 512 warm-up steps. For MDE adaptation, we also employ the Adam optimizer [6] with a learning rate of $1 \cdot 10^{-5}$. NeRF optimization is performed with a pixel batch size of 4,096, totaling 20K iterations. For $\mathcal{L}_{\text{seen}}$, we render a $64 \times 64$ patch, while for $\mathcal{L}_{\text{unseen}}$, we render a $128 \times 128$ patch with a stride of 3.

For the loss functions, we set the coefficients of $\mathcal{L}_{\text{seen}}$, $\mathcal{L}_{\text{MDE}}$, and $\mathcal{L}_{\text{reg}}$ as 0.01, 0.01, and 0.1, respectively. During the warm-up stage of 5,000 steps, the coefficient of $\mathcal{L}_{\text{unseen}}$ is initially set to 0 and then increased to 0.01 after 5,000 warm-up steps. For the first 1,000 steps, we employ the ranking loss [19] with a coefficient of 0.1, in addition to $\mathcal{L}_{\text{seen}}$. All experiments were conducted using a single NVIDIA GeForce RTX 3090. The training process takes approximately 3 hours.

### A.3  Training loss details

In the following, we describe a least-square alignment [18] used in loss functions for MDE prior distillation in detail. As described in the main paper, we use a scale-shift invariant loss [13] with patch-wise adjustment for depth consistency as follows:

$$\mathcal{L} = \sum_{I_i \in \mathcal{S}} \sum_{\mathbf{p} \in \mathcal{P}} \|(w_i D_i^*(\mathbf{p}) + q_i) - \bar{D}_i(\mathbf{p})\|, \tag{1}$$

where $w_i$ and $q_i$ are scale and shift values that align $D_i^*(\mathbf{p})$ to the absolute locations of $\bar{D}_i(\mathbf{p})$. In this loss function, to calculate $w_i$ and $q_i$, we following least-squares criterion [13]:

$$(w_i, q_i) = \arg \min_{w_i, q_i} \sum_{\mathbf{p} \in \mathcal{P}} \|(w_i D_i^*(\mathbf{p}) + q_i) - \bar{D}_i(\mathbf{p})\| \tag{2}$$

In other words, we can rewrite the above scheme as a closed problem. Let $\mathbf{h}_i = [w_i, q_i]^T$ and $\vec{D}_i(\mathbf{p}) = [D_i^*(\mathbf{p}), 1]^T$, then we can modify our problem as

$$\mathbf{h}_i^{\text{opt}} = \arg \min_{\mathbf{h}_i} \sum_{\mathbf{p} \in \mathcal{P}} (\vec{D}_i(\mathbf{p})^T \mathbf{h}_i - \bar{D}_i(\mathbf{p}))^2, \tag{3}$$

which can be solved as follows:

$$\mathbf{h}_i^{\text{opt}} = \left( \sum_{\mathbf{p} \in \mathcal{P}} \vec{D}_i(\mathbf{p}) \vec{D}_i(\mathbf{p})^T \right)^{-1} \left( \sum_{\mathbf{p} \in \mathcal{P}} \vec{D}_i(\mathbf{p}) \bar{D}_i(\mathbf{p}) \right) \tag{4}$$

### A.4 Baseline implementations

We directly use quantitative results reported in prior literature [16] for the comparison of Nerfing-MVS [17], DS-NeRF [3] and DDP-NeRF [14]. As the setting [16] requires out-of-domain priors, it should be noted that the results for DDP-NeRF are with out-of-domain priors. The results of DDP-NeRF with in-domain priors are 20.96, 0.737, and 0.236 for PSNR, SSIM, and LPIPS, respectively. However, we were unable to evaluate DDP-NeRF in the extreme settings of ScanNet and Tanks and Temples, as reliable COLMAP 3D points could not be obtained.

We utilized the authors' provided official implementations of RegNeRF [11] and $K$-planes [4], training one model for each scene using two different scenarios on the ScanNet [2] and Tanks and Temples [7] datasets. However, since there is no official code available for SCADE [16], we are unable to provide performance comparisons for this method.

# B  Datasets and Metrics

## B.1  Datasets

**ScanNet [2].**  We adhere to the few-shot protocol provided by DDP-NeRF [14] in our experimental setup. We noticed that the split contained major overlaps across the train and test sets, which makes the task easier compared to realistic few-shot settings where images exhibit minimal overlap. For this reason, we construct an extreme few-shot scenario, using only half of the training images while maintaining the same test set.

**Tanks and Temples [7].**  To test the robustness of our method in challenging real-world outdoor environments, we conduct further experiments on Tanks and Temples dataset, an real-world outdoor dataset acquired under drastic lighting effects and reflectances. As no existing protocols exist for a few-shot scenario for this dataset, we introduce a new split for the few-shot setting. We carefully selected 5 object-centric scenes —truck, francis, family, lighthouse, and ignatius— with inward-facing cameras. From each scene, we sample 10 training images that capture the overall geometry of the whole scene. For testing, we use one-eighth of the dataset as a test set, consisting every $8^{th}$ repeating image from the entire image set. We run COLMAP [15] on all images to obtain camera poses for NeRF training. However, for the lighthouse scene, which exhibits highly sensitive lighting and specular effects dependent on view pose, we manually preprocess the parts that contain these effects.

## B.2  Evaluation metrics

To evaluate the quality of novel view synthesis, following previous works [9], we measure PSNR, SSIM, and LPIPS. It is mentioned in $K$-planes that an implementation of SSIM from mip-NeRF [1] results in lower values than standard scikit-image implementation. For a fair comparison per dataset, we use the latter scikit-image SSIM implementation following the relevant prior work.

For the evaluation of the MDE module, we use 4 depth estimation metrics as follows:

- AbsRel: $\frac{1}{|\mathcal{I}|} \sum_{\mathbf{p} \in \mathcal{I}} \|\bar{D}(\mathbf{p}) - D^{\mathrm{GT}}(\mathbf{p})\| / D^{\mathrm{GT}}(\mathbf{p})$;
- SqRel: $\frac{1}{|\mathcal{I}|} \sum_{\mathbf{p} \in \mathcal{I}} \|\bar{D}(\mathbf{p}) - D^{\mathrm{GT}}(\mathbf{p})\|^2 / D^{\mathrm{GT}}(\mathbf{p})$;
- RMSE: $\sqrt{\frac{1}{|\mathcal{I}|} \sum_{\mathbf{p} \in \mathcal{I}} \|\bar{D}(\mathbf{p}) - D^{\mathrm{GT}}(\mathbf{p})\|^2}$;
- RMSE log: $\sqrt{\frac{1}{|\mathcal{I}|} \sum_{\mathbf{p} \in \mathcal{I}} \|\log \bar{D}(\mathbf{p}) - \log D^{\mathrm{GT}}\|^2}$;

where $\mathbf{p}$ is a pixel in the image $\mathcal{I}$ and $D^{\mathrm{GT}}$ is ground truth depth map. In addition, following [20], we use single scaling factor $s$ for each scene which is obtained by

$$s = \frac{1}{N} \sum_{I_i \in \mathcal{S}} (\mathrm{median}(D_i^{\mathrm{GT}} / \bar{D}_i)), \tag{5}$$

rather than fit each frame to ground truth, to evaluate view consistency of MDE models. Here, $\mathcal{S}$ denotes set of images from single scene.

# C  Additional Analysis

## C.1  Comparison of patch- and image-level scale-shift adjustment

We provide additional analysis and visualization results regarding the patch-wise scale and shift adjustment. In Fig.1 and Fig.2, we present error maps showing the discrepancies between the ground truth sensor depth and the predicted depth. Additionally, in Fig.3, we present qualitative results of rendered color and depth using each fitting method. It is important to note that in the image-level fitting scheme, a single set of scale and shift values is computed for an entire depth map. Conversely, in our patch-level fitting method, scale and shift values are calculated individually for each $80 \times 80$ patch within the depth map. The error map clearly demonstrates the significant reduction in misalignment errors achieved by our patch-level fitting method compared to the image-level fitting approach.

For the comparison of image-level and patch-level fitting provided in the Fig.2 and Tab.5 of the main paper, we set the scale and shift as learnable parameters per image for image-level fitting and conduct patch-wise scale-shift invariant loss for patch-level fitting. This comparison is conducted only with $\mathcal{L}_{\text{seen}}$ given and results with patch-level fitting show better performance compared to image-level fitting. The difference between the two methods is especially distinguished in rendered depth maps of these two settings, in that patch-level fitting lets NeRF learn depth more accurately.

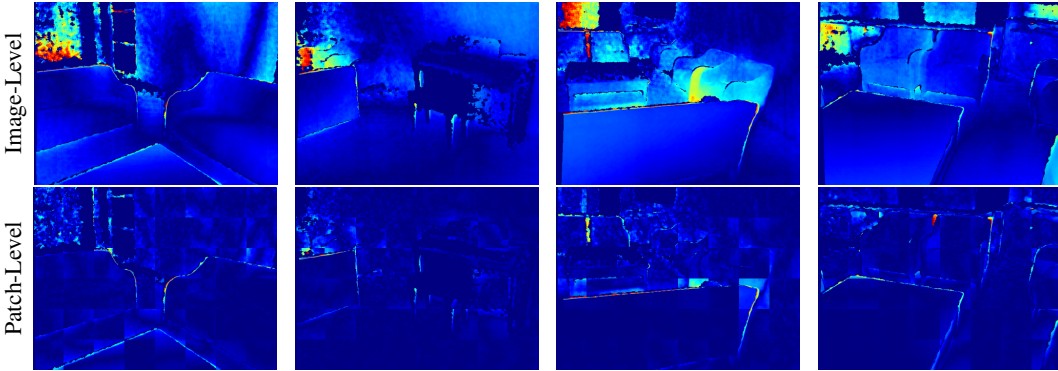

Figure 1: **Error map visualization of image-level and patch-wise scale and shift adjustment:** relative depth map in various viewpoints is fitted in two ways, image-level fitting (first row) and patch-level fitting (second row).

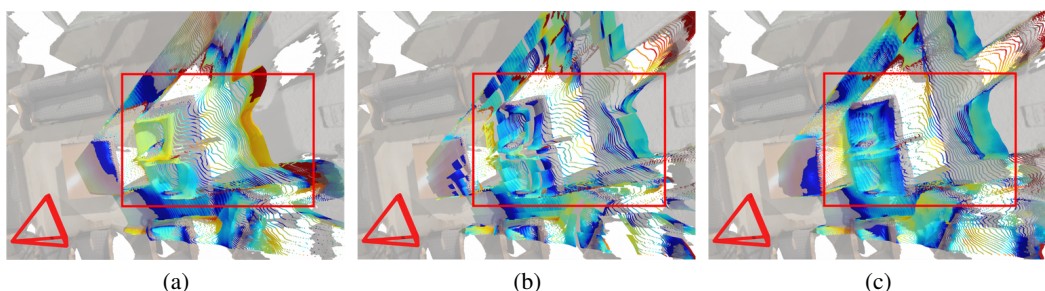

(a)                         (b)                         (c)

Figure 2: **3D Visualization of the error map of MDE and NeRF:** (a) monocular depth with image-level adjustment, (b) monocular depth with patch-level adjustment, and (c) rendered depth by NeRF trained with patch-level adjustment. Depth from the input image of the viewpoint stated as red camera is adjusted in each ways and unprojected into 3D space. Error of each point cloud of a room is visualized from the bird's eye view. This is done with jet color coding, so that red color means large error and blue color means small error. The proposed patch-wise adjustment helps to minimize the errors caused by inconsistency in depth differences among objects.

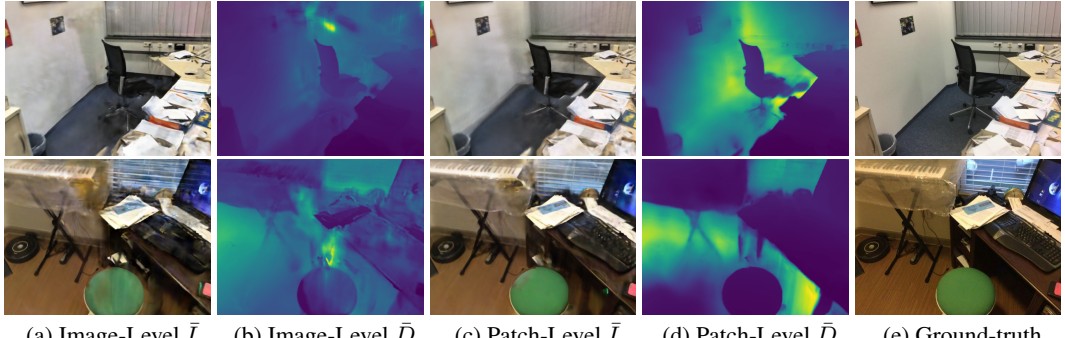

| (a) Image-Level $\bar{I}$ | (b) Image-Level $\bar{D}$ | (c) Patch-Level $\bar{I}$ | (d) Patch-Level $\bar{D}$ | (e) Ground-truth |

Figure 3: **Comparison of patch- and image-level scale-shift adjustment.** Rendered color and depth from NeRF with (a-b) image-level scale and shift adjustment and (c-d) patch-level scale and shift adjustment.

## C.2 Confidence modeling

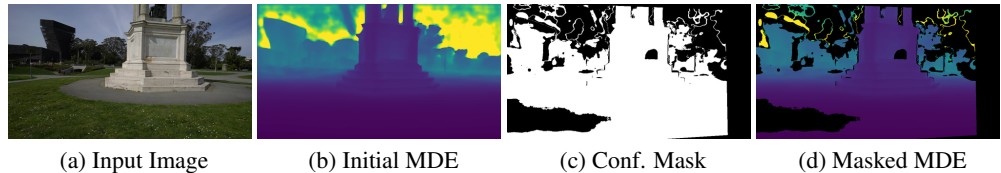

| (a) Input Image | (b) Initial MDE | (c) Conf. Mask | (d) Masked MDE |

Figure 4: **Comparions on MDE depth map with and without confidence masking.** the initial MDE depth map predicted is filtered through mask from our confidence modeling.

In Fig. 4, we demonstrate the effectiveness of our confidence modeling which effectively eliminates inaccurate information present in depth maps from both NeRF and the MDE network through leveraging multi-view consistency of NeRF. MDE depth from the input image contains errors, which can be filtered out by verifying consistency with depth from NeRF's other viewpoint. Likewise, the error of MDE depth from unseen viewpoint can be filtered through consistency check with MDE depth from the seen viewpoint.

## C.3 Ablation of MDE baselines

Table 1: **Ablation study on MDE baseline.**

| Components | PSNR↑ | SSIM↑ | LPIPS↓ |
|---|---|---|---|
| DäRF with LeReS [19] | 21.31 | 0.757 | 0.343 |
| DäRF with MiDaS [13] | 21.48 | 0.758 | 0.337 |
| DäRF with DPT [12] | 21.58 | 0.765 | 0.325 |

We conduct an ablation on the Monocular Depth Estimation (MDE) network to assess its impact on our methodology. Considering the recent advancements [13, 12] in MDE models that shows strong generalization power for depth estimation in unseen images, we replace our MDE network with state-of-the-art models such as LeReS, MiDaS, and DPT. The results in Tab. 1 show that our method shows consistent performance across different baselines.

# D  Camera Visualization of proposed Few-shot setting

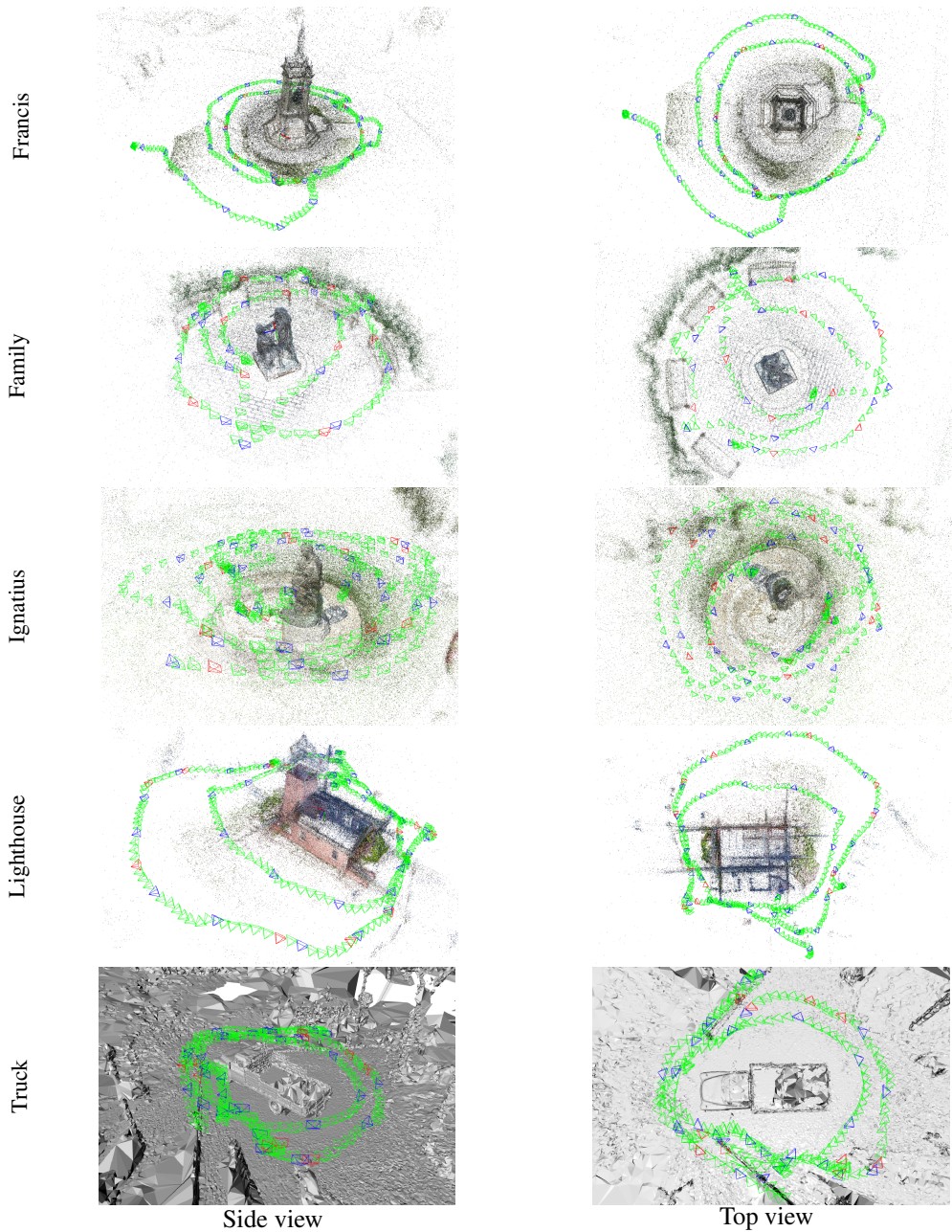

Side view                    Top view

Figure 5: **Camera visualization of Tanks and Temples dataset.** Green cameras mean all sets of images provided, and red and blue cameras mean train and test sets. As shown here, red cameras, i.e., train set, are a very small fraction of the camera set with little overlapping. On the other hand, blue cameras, i.e., test set, cover various locations, distributed in various positions of the scene.

# E   Additional Qualitative Results

In this section, we show additional qualitative comparisons in Fig. 6, Fig. 7, Fig. 8, Fig. 9, Fig. 10, and Fig. 11 for ScanNet [2] dataset in two different settings and in Fig. 12, Fig. 13, Fig. 14, Fig. 15, and Fig. 16 for Tanks and Temples [7] dataset.

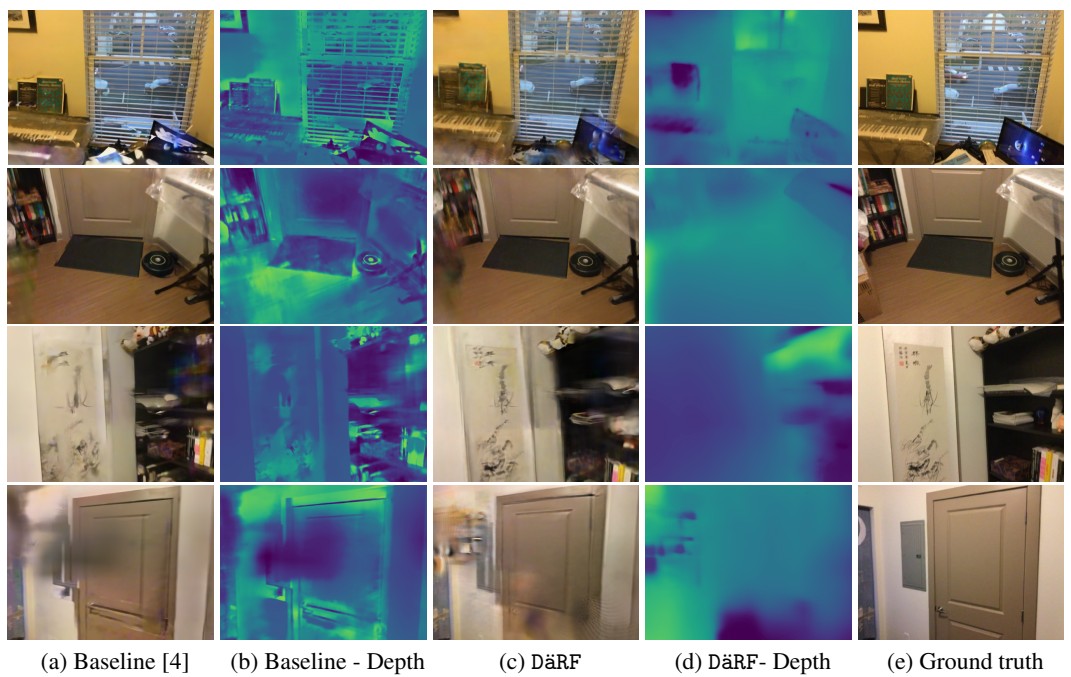

(a) Baseline [4]   (b) Baseline - Depth   (c) DäRF   (d) DäRF- Depth   (e) Ground truth

Figure 6: **Qualitative results on Scan 0710 of ScanNet [2] with 9 - 10 input views**.

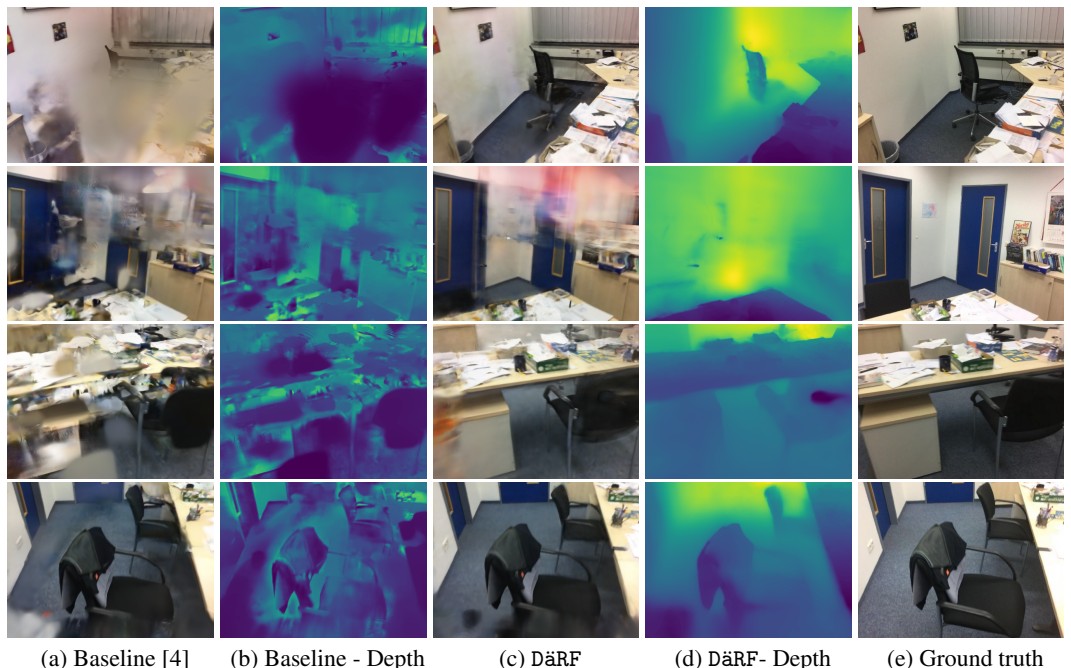

(a) Baseline [4]   (b) Baseline - Depth   (c) DäRF   (d) DäRF- Depth   (e) Ground truth

Figure 7: **Qualitative results on Scan 0758 of ScanNet [2] with 9 - 10 input views**.

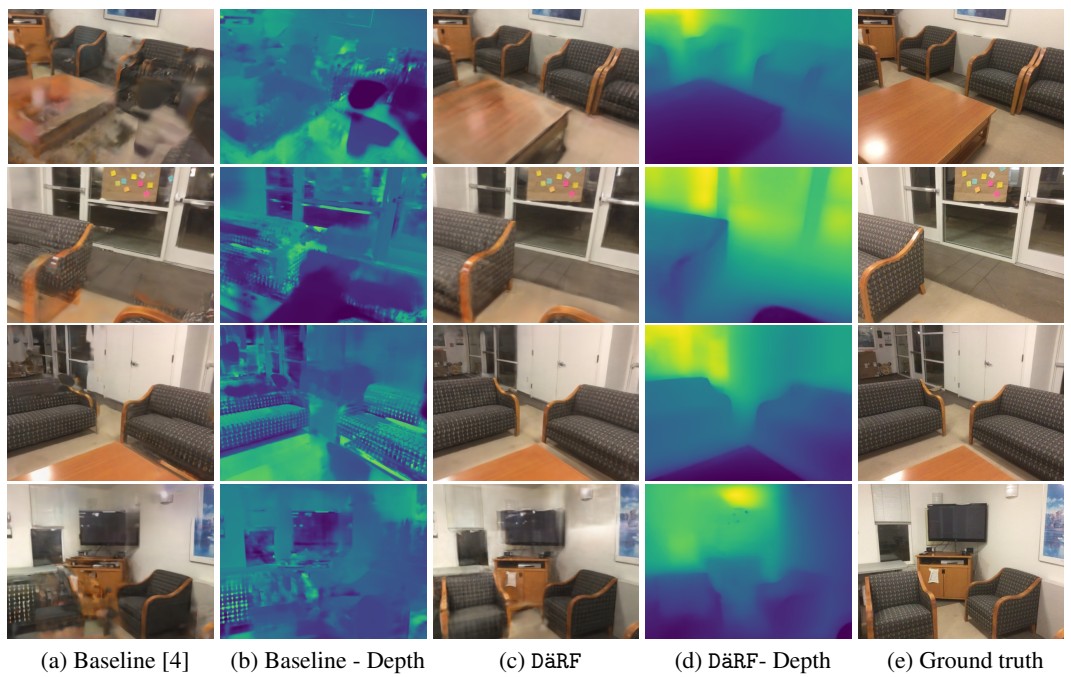

(a) Baseline [4]     (b) Baseline - Depth     (c) DäRF     (d) DäRF- Depth     (e) Ground truth

Figure 8: **Qualitative results on Scan 0781 of ScanNet [2] with 9 - 10 input views**.

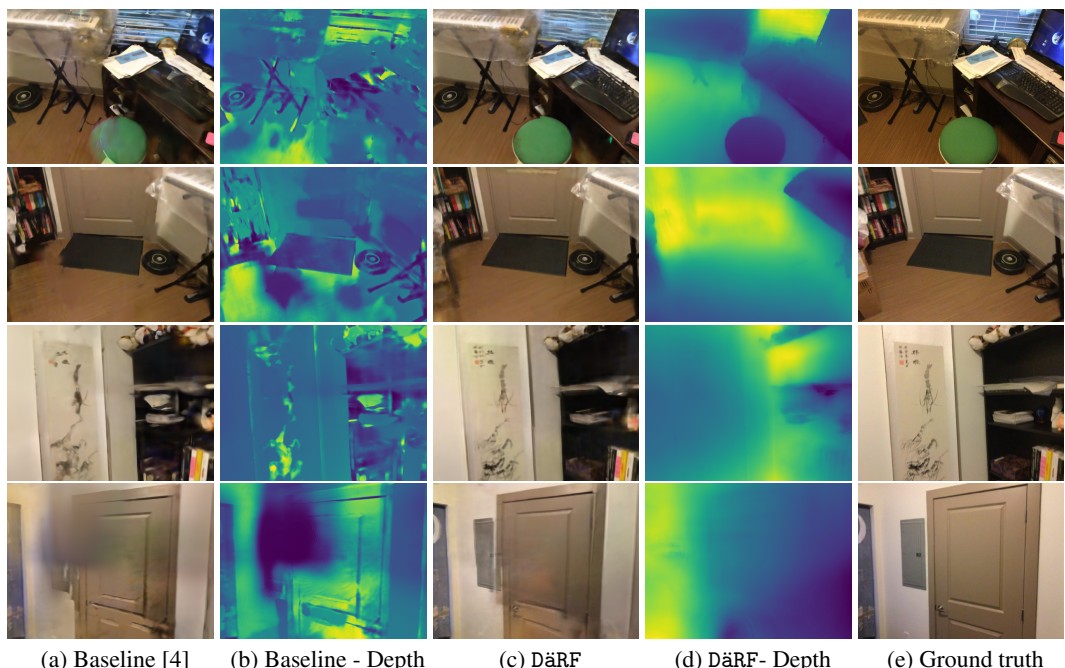

(a) Baseline [4]     (b) Baseline - Depth     (c) DäRF     (d) DäRF- Depth     (e) Ground truth

Figure 9: **Qualitative results on Scan 0710 of ScanNet [2] with 18 - 20 input views**.

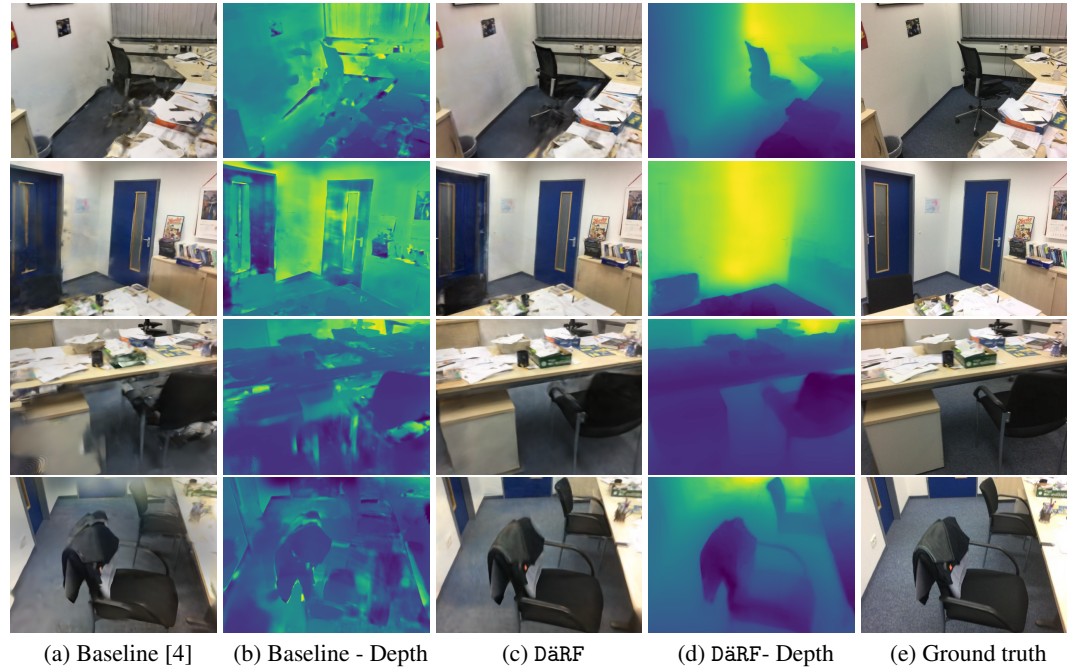

(a) Baseline [4]    (b) Baseline - Depth    (c) DäRF    (d) DäRF- Depth    (e) Ground truth

Figure 10: **Qualitative results on Scan 0758 of ScanNet [2] with 18 - 20 input views**.

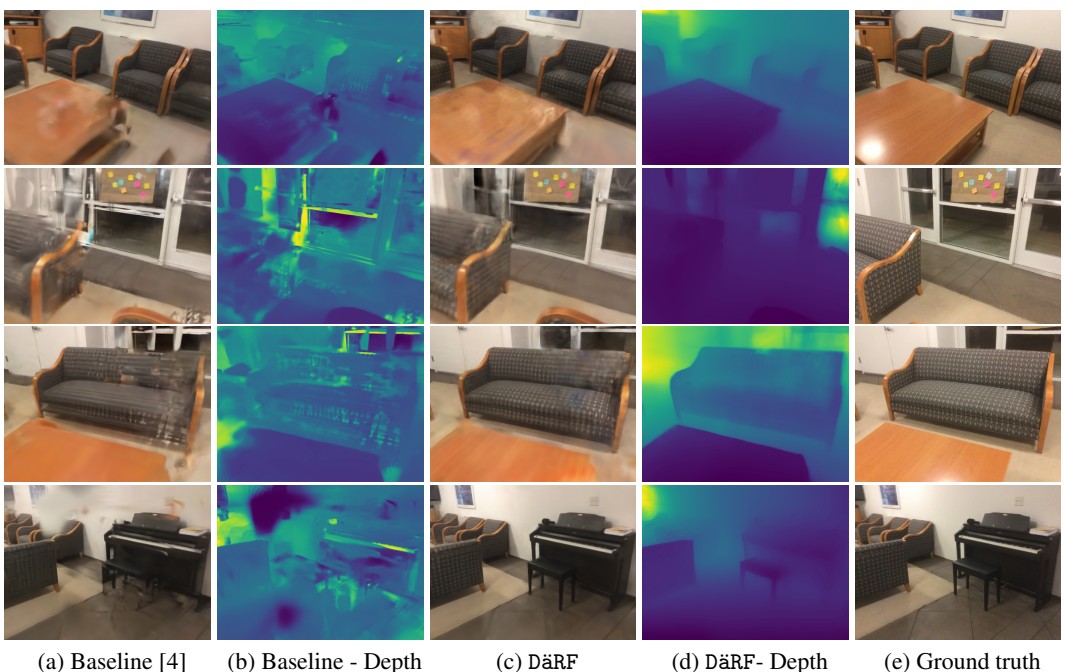

(a) Baseline [4]    (b) Baseline - Depth    (c) DäRF    (d) DäRF- Depth    (e) Ground truth

Figure 11: **Qualitative results on Scan 0781 of ScanNet [2] with 18 - 20 input views**.

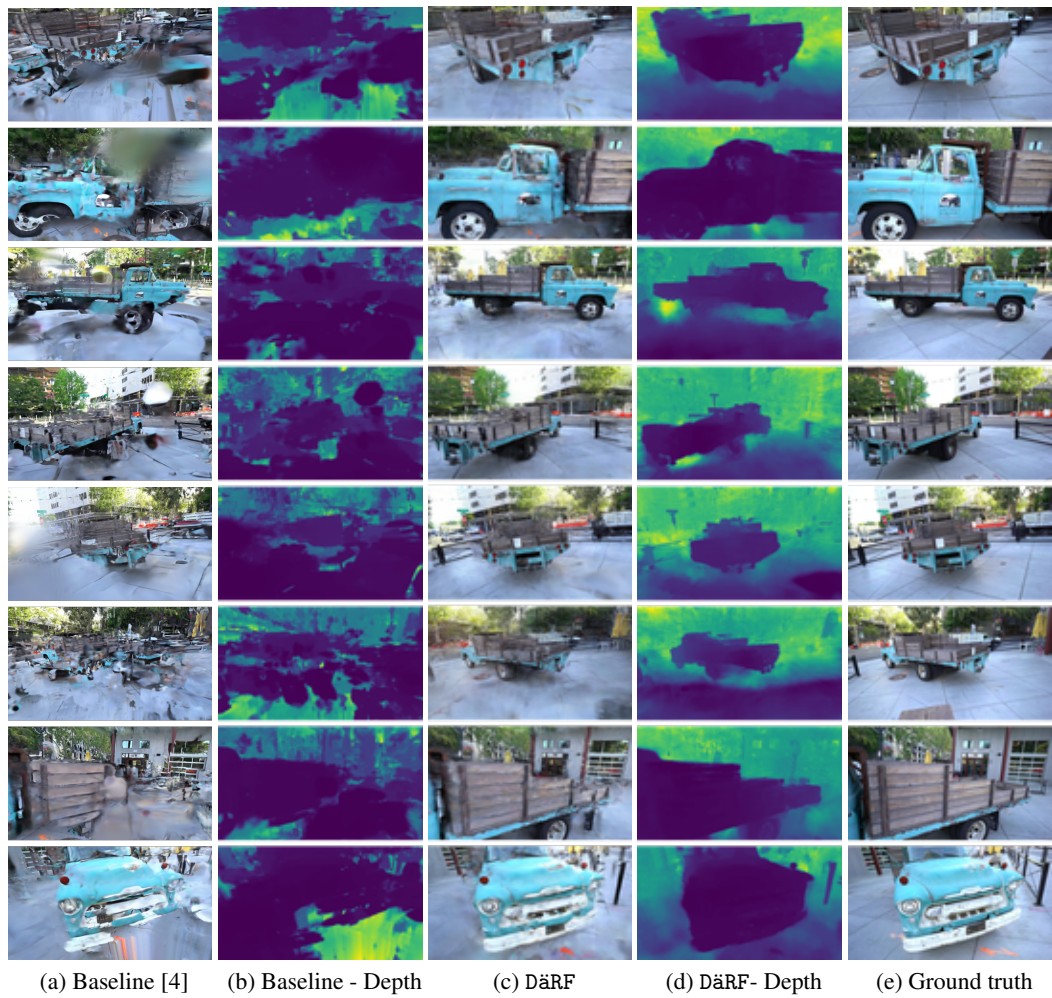

(a) Baseline [4]    (b) Baseline - Depth    (c) DäRF    (d) DäRF- Depth    (e) Ground truth

Figure 12: **Qualitative results on truck scene of Tanks and Temples [7] with 10 input views**.

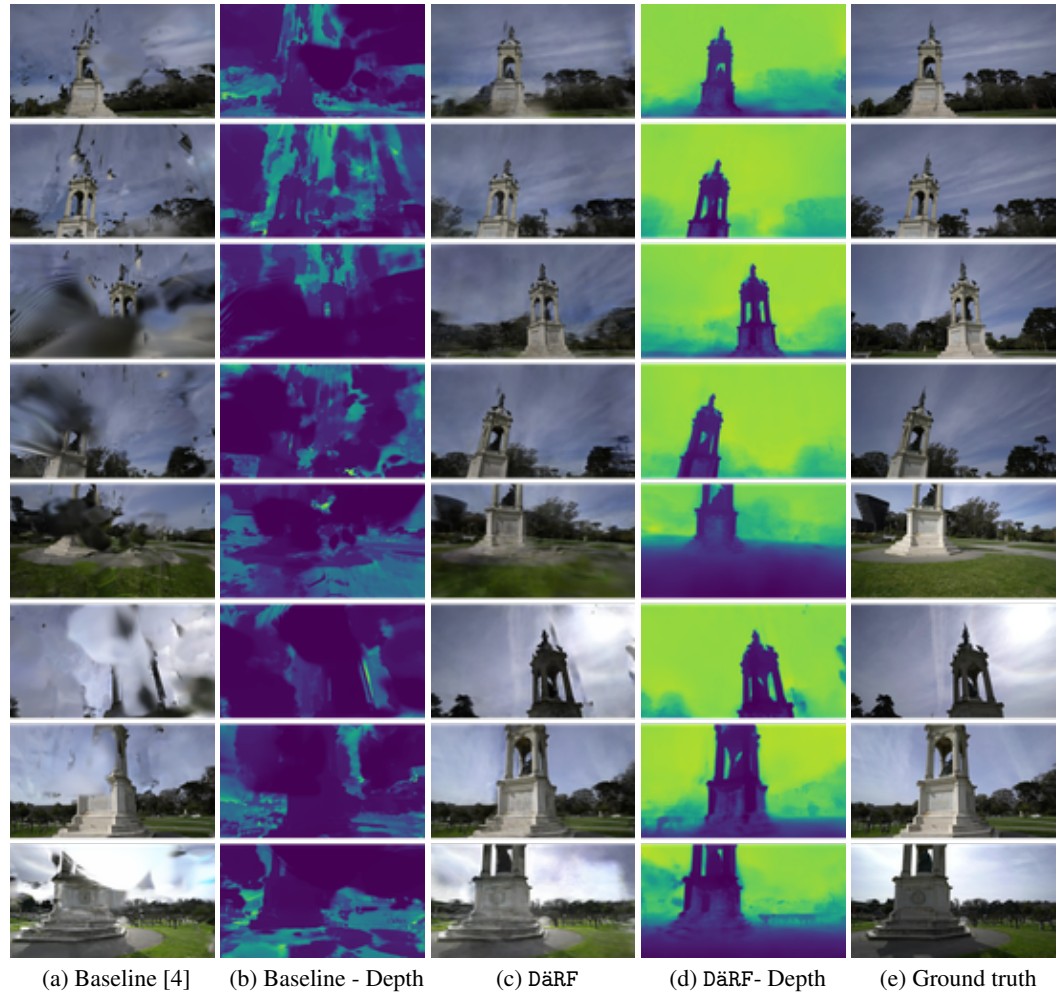

(a) Baseline [4]  (b) Baseline - Depth  (c) DäRF  (d) DäRF- Depth  (e) Ground truth

Figure 13: **Qualitative results on francis scene of Tanks and Temples [7] with 10 input views**.

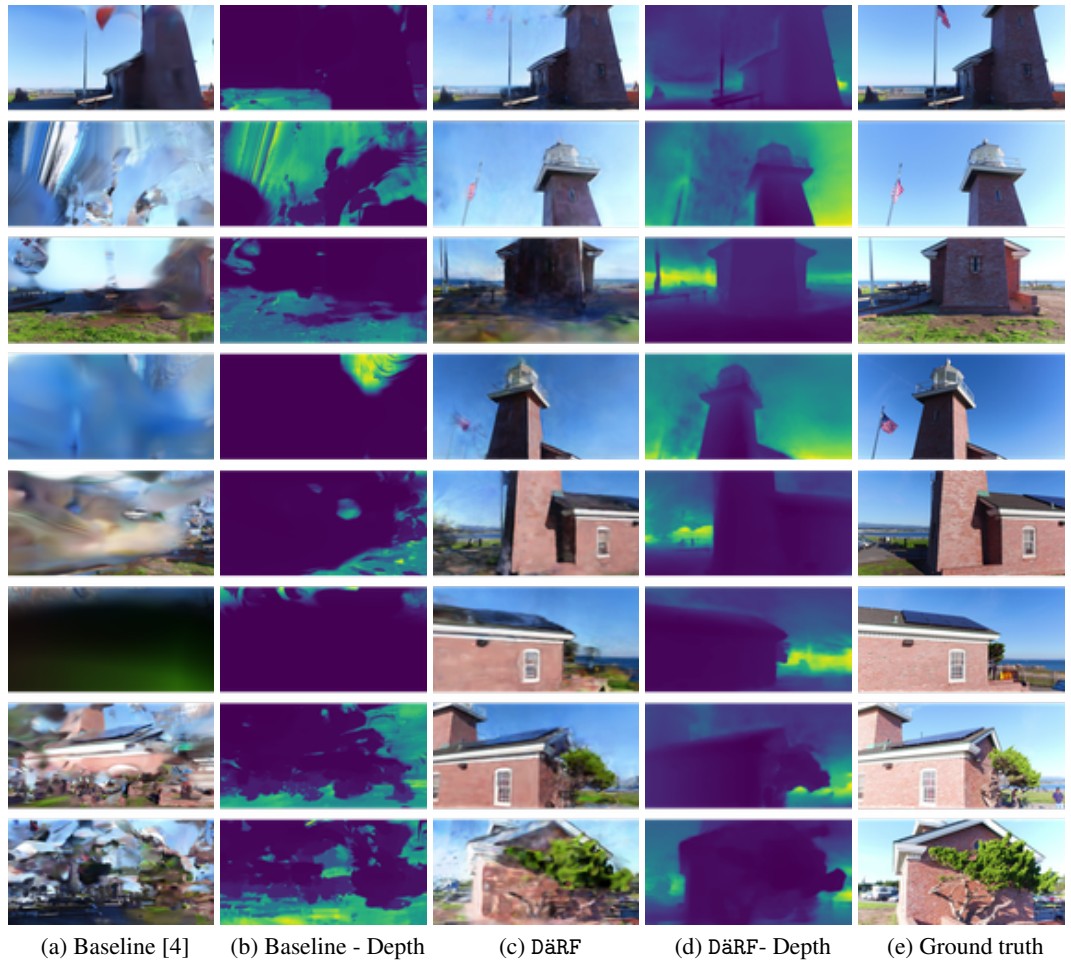

(a) Baseline [4]    (b) Baseline - Depth    (c) DäRF    (d) DäRF- Depth    (e) Ground truth

Figure 14: **Qualitative results on lighthouse scene of Tanks and Temples [7] with 10 input views**.

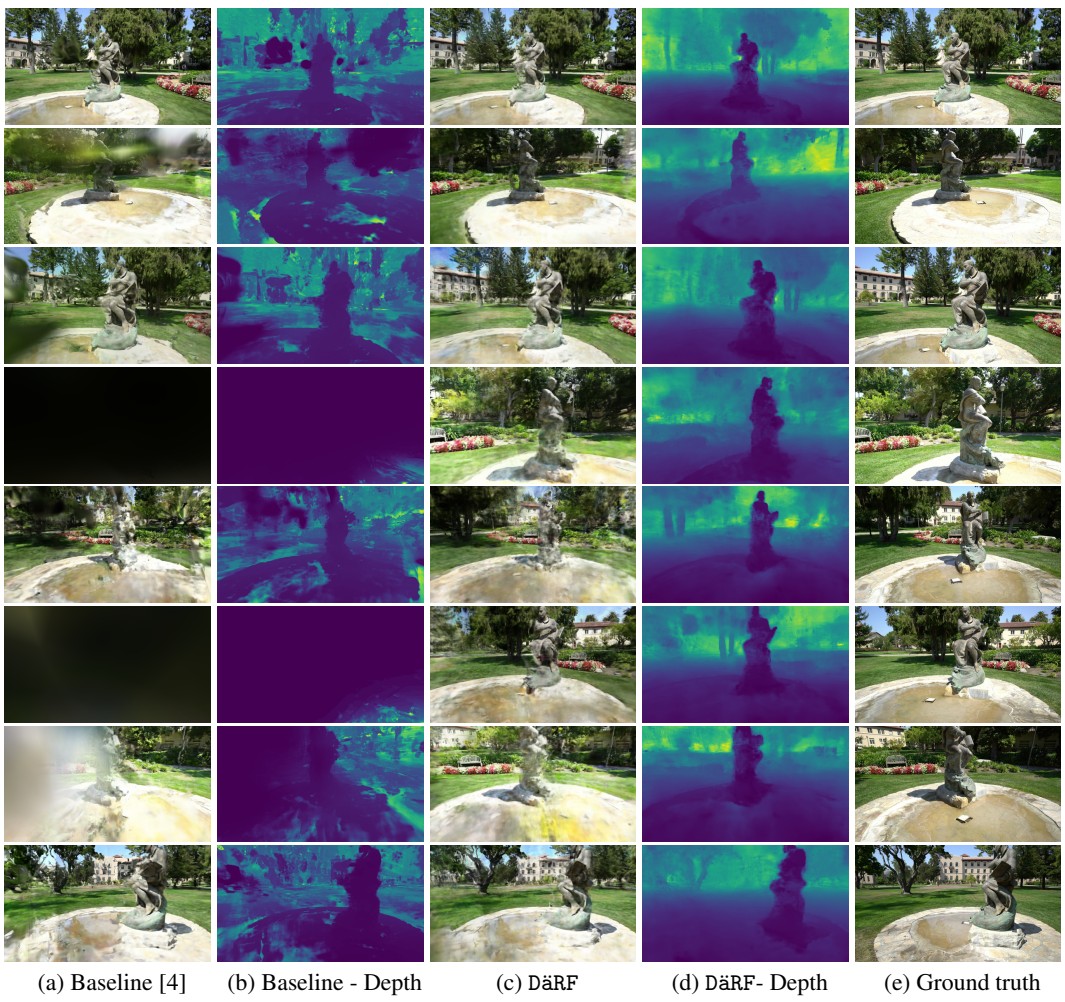

(a) Baseline [4]    (b) Baseline - Depth    (c) DäRF    (d) DäRF- Depth    (e) Ground truth

Figure 15: **Qualitative results on ignatius scene of Tanks and Temples [7] with 10 input views**.

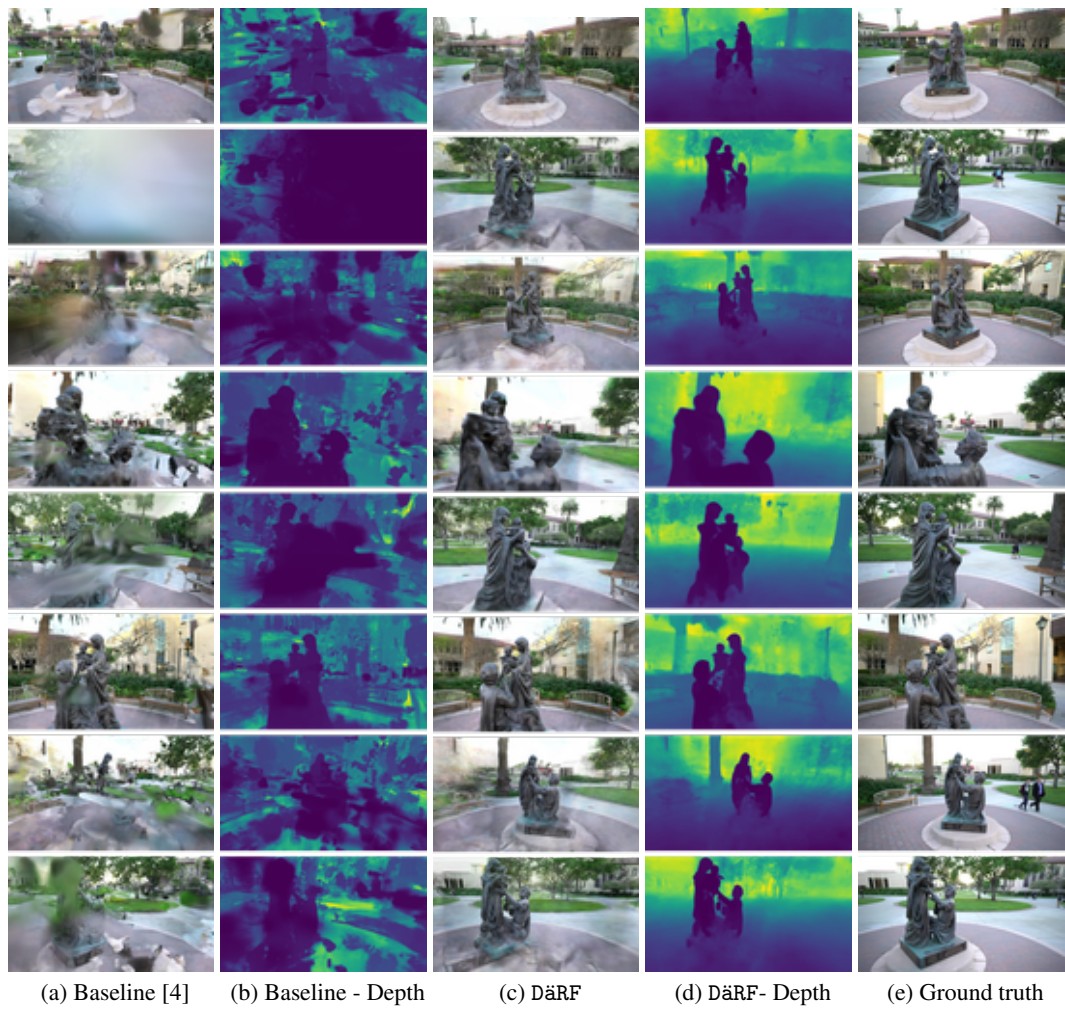

(a) Baseline [4]   (b) Baseline - Depth   (c) DäRF   (d) DäRF- Depth   (e) Ground truth

Figure 16: **Qualitative results on family scene of Tanks and Temples [7] with 10 input views**.

# F   Limitations and Future Works

While our method shows powerful performance quantitatively, its limitations can be noticed in its qualitative results above, where it struggles to reconstruct the fine-grained details present in ground truth images. Also, our usage of depth supervision from various viewpoints does not get rid of the artifacts completely: some artifacts that cloud the space between objects and the camera, are reduced yet still visible in rendering of unseen viewpoints.

These limitations may be attributed to fundamental limitations in the few-shot NeRF setting [5], where fine-grained details are often occluded from one viewpoint to another due to an extreme lack of input images, preventing faithful geometric reconstruction of details. Also, since the seen viewpoints view a comparatively small portion of the entire scene, there inevitably occur artifacts in the unseen viewpoint as some depths cannot be perfectly determined from given input information.

# G   Broader Impacts

Our work achieves robust optimization and rendering of NeRF under sparse view scenarios, drastically reducing the number of viewpoints required for NeRF and bringing NeRF closer to real-life applications such as augmented reality, 3D reconstruction, and robotics. Our extension of few-shot NeRF to a real-world setting with the usage of monocular depth estimation networks also would enable NeRF optimization under various real-life lighting conditions and specular surfaces due to its increased robustness and generalization power.