# OpenReview forum: "DäRF: Boosting Radiance Fields from Sparse Input Views with Monocular Depth Adaptation"
_NeurIPS.cc/2023/Conference — NeurIPS 2023 poster_

### Official Review · Reviewer_VhkU · 2023-06-29

**Soundness:** 2 fair
**Presentation:** 2 fair
**Contribution:** 2 fair
**Rating:** 6
**Confidence:** 4

**Summary:**

The paper proposes a method to better use monocular depth in a few-shot NeRF setup. There are mainly two technical contributions to me:
1. Applying mono depth constraint to unseen view;
2. Un-distorting (scale and shift) monocular depth in a per-patch manner, rather than per-image.

---
**After rebuttal**: I have read authors' rebuttal and it addresses my concerns.

**Strengths:**

1. The motivation makes a lot of sense. Monocular depth estimation suffers from various distortions, and un-distorting it with a global scale and shift per image is only a rough approximation. Exploring better ways to un-distort monocular depth is definitely valid motivation.
2. It’s interesting to see that monocular depth networks actually perform decently on rendered images from half-trained NeRF models.

**Weaknesses:**

General
1. L118-L120: I disagree with this definition of few-shot, i.e. $| S < 20 |$. The definition should be based on view-angle and scene coverage, not on an absolute number of images. For example, in LLFF, a forward-facing scene, NeRF can be trained very well for $| S < 20 |$, because 20 images cover the simple forward-facing scenes very well. In contrast, 50 images might be challenging for 360-degree scenes.
2. Following point 1, how do the few-shot views cover the evaluated scenes? It would be good to have a top-view visualisation of selected cameras' positions and orientations.
3. How many patches are used in training? From the supp mat I can see the patch size is 64x64, i.e. 4096 rays. This means there is only one patch used during training? In this case, modelling patch-wise scale/shift is same as image-wise scale/shift?

Writing
1. Figure 1 caption: unclear, seems like should be … by _applying_ pretrained MDE … (missing a word _applying_)
2. Figure 2:
    1. what is the input RGB and its monocular depth? I can see this is a top-view image of back-projected point clouds, but I cannot imagine what’s the input.
    2. What is colour coding?
    3. I don’t see why Fig 2b is better than Fig 2a from this image. I understand the motivation and it is supposed to be better when un-distort with patch-wise scale/shift, but I cannot see that from this visualisation.
3. Sec 4.1 is kind of repeating intro.
4. Symbol $M_l$ is undefined. I suppose it’s similar to $M_i$, but projected in $l \rightarrow i$ direction?
5. Symbol $s_i, t_i$ are undefined.

**Questions:**

See the weaknesses section.

**Limitations:**

Yes.

---

> ### Author Rebuttal · Authors · 2023-08-10
>
> > **Q1. Specific definition of few-shot setting and its visualization**
> >
>
> We agree with your statement that the precise definition of few-shot cannot be defined naively as |S<20|, described in our L118-L120. As you have said, its definition should be based on view-angle and general coverage of the scene. However, due to this very reason, it is difficult to precisely define the minimum number of viewpoints required for few-shot reconstruction, as that number will vary scene by scene according to each scene’s specific shape, geometry, and occlusions.
>
> In this light, current few-shot NeRF works [1,2,3,4] assume extreme cases where i) the known viewpoints overlap with each other minimally, ii) all directions of the scene are viewed at least once so that there are no regions of the scene where no information is given. Such extreme scenarios cause original NeRF models to struggle heavily with geometric reasoning so that they cannot perform proper reconstruction, which necessitates the introduction of few-shot regularization methods.
>
> Considering these criteria and the general characteristics of the scenes in the dataset, they decide upon a certain number of viewpoints that result in such extreme setting per dataset. For example, the forward-facing *LLFF dataset* selects equally distant 3 views for few-shot setting, and *NeRF-synthetic dataset*, despite its 360 degree viewpoint, also assumes 3 views as its few-shot setting considering its simplicity.
>
> In this manner, we constructed few-shot train/test sets of Tanks and Temples dataset, where few-shot standard does not yet exist, considering its complexity, 360 degrees, real-world dataset. Following your constructive suggestion, we visualize how these few viewpoints cover our scenes in a minimal yet sufficient manner in Figure 1 of our attached pdf file. We believe this figure will be beneficial in justifying our few-shot setting to the readers, and we promise to include it in the final version of our paper. Thank you for the helpful comment.
>
> > **Q2. Number of patches used in training**
> >
>
> We clarify that we sample each patch per iteration, and each patch’s scale/shift value does not equate to global image-wise scale/shift values.
>
> More specifically, we sample a 64 by 64 patch and a 128 by 128 patch per iteration, but these patches are randomly sampled from a random seen pose and a random unseen pose each other, and it is this corresponding patch-wise region that we regularize with predicted MDE depth. In every iteration, both viewpoint and patch locations are randomly sampled a new, and the patch-wise scale-shift values are modeled accordingly. Since the viewpoints and patches are sampled uniformly, our regularization signal gives coverage of every possible region of the scene throughout the course of optimization. In this way, the local supervision signal influences all regions of the scene as optimization progresses.
>
> > **Q3. Details of Figure 2**
> >
>
> Thank you for your constructive comment. We agree that Figure 2 is difficult to recognize at first sight, and we promise to fix it.
>
> Our intention for Figure 2 was to visualize our point that a single scale-shift value cannot perfectly fit into ground truth geometry due to the ambiguity of distances between object instances. For this reason, we have visualized **error** of a predicted point cloud of a room from the bird’s eye view to show that patch-wise fitting most accurately fits point cloud to ground truth geometry. This is done with jet color coding, so that red color means large error and blue color means small error. This point cloud is projected depth from the input image from the viewpoint stated as red camera.
>
> Following your comment, we promise to either replace it with Figure 1 of our supplementary material (which describes the same phenomenon more clearly) or reinforce it with an additional caption that explains it in a more detailed manner.
>
> > **Q4. Clarifications in writing**
> >
>
> Thank you for pointing these out. We promise to make a modification to the caption of Figure 1 and Section 4.1 in the final version of our paper.
>
> Also, we clarify that $M_l$ shares a similar definition with $M_i$, which we define as a confidence mask for seen viewpoint $i$ - therefore, $M_l$ corresponds to a confidence mask for unseen viewpoint $l$. Also, the notation $s_i,t_i$ in supplementary materials is a notation mistake: it is supposed to be $w_i, q_i$. Thank you for the careful reading and revision of our paper: we promise to clarify & revise these notations accordingly in the final version of our paper.
>
> [1] Jain et al., Putting NeRF on a Diet: Semantically Consistent Few-Shot View Synthesis, ICCV 2021.
>
> [2] Kim et al., InfoNeRF: Ray Entropy Minimization for Few-Shot Neural Volume Rendering, CVPR 2022.
>
> [3] Niemeyer et al., RegNeRF: Regularizing Neural Radiance Fields for View Synthesis from Sparse Inputs, CVPR 2022.
>
> [4] Yang et al., FreeNeRF: Improving Few-shot Neural Rendering with Free Frequency Regularization, CVPR 2023.

---

> > ### Comment · Reviewer_VhkU · 2023-08-16
> > **Thanks for the additional results and detailed response**
> >
> > The idea of adapting monocular depth to novel view images that are still under training is interesting to me. The new response addresses most of my questions and concerns, also Fig. 1 in the rebuttal pdf is very helpful so I suggest including it in supp mat later. Overall I'll raise my rating in the final review.
> >
> > Just one quick clarification regarding the patch size: I understand the patches are randomly sampled in each iteration, but is the patch cropped from an image or is a $64 \times 64$ patch formed by randomly sampling 4096 pixels from an image and reshaping to $64 \times 64$?

---

> > > ### Author Response · Authors · 2023-08-17
> > > **Thanks and more clarification**
> > >
> > > Dear Reviewer VhkU,
> > >
> > > Thank you for your encouraging feedback and for increasing your score. We will add Fig. 1 in the rebuttal pdf in the final version of paper.
> > >
> > > For further clarification, a 64×64 patch is cropped in a patch-wise manner from an image of a randomly selected camera pose. Please let us know if you have remaining questions or concerns.
> > >
> > > Regards, Authors

---

> > > > ### Comment · Reviewer_VhkU · 2023-08-17
> > > > **thanks**
> > > >
> > > > Thanks a lot for the clarification. I don't have any other questions. Best, reviewer.

---

### Official Review · Reviewer_Ndsp · 2023-07-03

**Soundness:** 3 good
**Presentation:** 2 fair
**Contribution:** 2 fair
**Rating:** 5
**Confidence:** 4

**Summary:**

The paper tackles the problem of sparse view NeRF reconstruction by leveraging on monocular depth estimation networks as a prior. The main difference between DaRF and existing work is it also computes for a depth loss on unseen views, in contrast to prior works that only constrain depth on the training views. Moreover, they also adapt, i.e. fine tune, the monocular depth estimation network to agree with the depth produced by the NeRF. They also introduce a confidence term that determines which pixels to use the depth loss on. They conduct experiments on Scannet and Tanks and Temples dataset.

**Strengths:**

The paper proposes to use monocular depth estimation as a prior to constrain NeRF optimization under the sparse view regime. Different from existing works, DaRF found that they can also use the monocular depth prior on unseen views, and even on NeRF rendered depth on early stage of the training, the monocular depth network is able to produce reasonable and cleaner depth maps that can supervise and constrain the NeRF loss (as shown in Figure 3). This is the main contribution and finding of this submission. The authors also showed an ablation study to justify the different components that were introduced.

**Weaknesses:**

The concerns I have for the paper is whether 1) the contribution introduced is enough for paper acceptance, and 2) how it positions itself and it claims in contrast to existing works. For the first point, the main contribution and distinction it has is the finding that the monocular depth prior can also be used to constrain unseen viewpoints. As shown in Figure 3, even on noisy NeRF renderings, the MDE prior can produce clean depth maps. An add-on to this is the slight improvement in results (as shown in the ablation study) by also fine-tuning the MDE network as NeRF is being optimized. However, the way the paper was written and motivated is the ambiguity in monocular depth estimation -- as written in the intro, Sec 4.1 and Sec 4.3, which is the main premise and story of the existing (though somewhat concurrent) work SCADE. The paper did cite SCADE, but however only mentions the difference that DaRF is able to train on unseen views. Differentiating with existing work and clarifying contributions can be improved for both paper presentation as well as the claims made for the submission. Now, the concern and question is whether 1) being the contribution of DaRF is enough as a contribution to meet the bar of Neurips. And for this reason I am giving an initial rating of borderline reject and would like to hear from the other reviewers.

**Questions:**

1. How are the viewpoints selected for the unseen views that was used for L_{unseen}? How many unseen viewpoints are selected and how is the loss balanced with the seen training views? This in my opinion is an important detail to justify and back its main claim.
2. What is the intuition behind using two terms in Eq 6? Specifically, wouldn't the second addend term be enough? It seems redundant and not too intuitive.
3. What dataset was used for the Ablation study (Table 3)? Is it the average across all scenes in Scannet? Figure 8 shows one example from scene781 of scannet.

**Limitations:**

The authors included the limitations in their supplementary material. However, according to the checklist, they claim to include the error bars. However, I don't think I saw error bars reported in the submission.

---

> ### Author Rebuttal · Authors · 2023-08-10
>
> > **Q1. Contribution of DaRF / comparison with SCADE**
> >
>
> Thank you for your comment. While SCADE [1] does have a similar setting to ours (leveraging MDE for few-shot NeRF reconstruction), we would like to emphasize that there are **critical differences** that position our work orthogonal to SCADE.
>
> There is a fundamental difference between our work and SCADE’s approach toward handling ambiguity present within MDE: our work proposes an online, scene-specific adaptation of MDE which **directly adapts it to predict absolute, canonical geometry** in accordance with few-shot NeRF. Our novel local scale/shift alignment also aids this process. On the other hand, SCADE **injects uncertainty into MDE** through additional pretraining so that canonical geometry can be estimated through probabilistic modeling between multiple modes of estimated depths. So, while the ultimate goal regarding our MDEs may be similar (ambiguity removal), the core idea behind them are more or less opposite: our approach **directly removes ambiguity** present in MDE by leveraging canonical geometry captured by few-shot NeRF, while SCADE handles MDE through ***embracing its uncertainties and ambiguities** (as stated in its introduction)* and solving it with a probabilistic approach.
>
> Through this key difference, our approach attains many advantages over SCADE:
>
> 1. First, our approach does not require **additional pre-training of MDE** that SCADE requires. Because our online MDE adaptation requires no additional dataset nor training stages that SCADE relied upon, but enables a direct usage of an off-the-shelf MDE for few-shot NeRF, it is simpler, faster, and cheaper in aspects of training time and computation cost.
> 2. Second, this approach **bypasses SCADE’s process of predicting multiple (20) depths of each input image** for probabilistic modeling, which also costs more training time and computational cost. Since our MDE is adapted in a more precise, scene-specific manner, it only needs to predict a single depth map that can be analytically fitted to NeRF through our **novel patchwise scale-shift fitting**, which is also our major contribution.
> 3. Our usage of MDE at **unseen viewpoints** for regularization effect and artifact removal, a contribution you pointed out, is, therefore, a natural extension that is possible due to our adapted MDE’s enhanced ability to predict canonical geometry, as noted in Strength 1 of reviewer KpuH.
> 4. Finally, we point out that this cheaper and faster approach achieves **higher performance** than SCADE.
>
> To summarize, despite having similar objectives, our methodology introduces an orthogonal strategy of leveraging MDE through online training of NeRF for per-scene adaptation. This enables our model to achieve higher performance without SCADE’s MDE pretraining, multiple per-depth predictions, or probabilistic modeling. However, as the two methods are orthogonal, both methods may be able to be used in conjunction for even higher performance and more efficient few-shot NeRF regularization, which would be an exciting direction to expand our research. We are grateful for your comment and suggestion.
>
> > **Q2. Details regarding unseen view selection and loss**
> >
>
> We generate an unseen camera pose every iteration, adding noise value sampled between an interval of [-6, +6] to the original Euler rotation angles of randomly selected input view pose. As mentioned in training details in our supplementary materials, the weights for the seen and unseen viewpoint loss are the same.
>
> > **Q3. Regarding intuition behind Eq6 / redundant terms**
> >
>
> We clarify that our methodology revolves around the idea of **adapting MDE toward predicting a scene-specific absolute geometry**, which is achieved by the first addend term: this first term forces itself MDE to adapt towards multiview consistency so that its ill-posed nature is reduced and its initial **global depth prediction** grows to be more in accordance with the absolute geometry captured by NeRF.
>
> In contrast, the second addend term, which takes into account patchwise scale-shift fitting, is designed to aid the modeling of **fine, detailed, local geometry** which the model has difficulty modeling without such local fitting.
>
> We emphasize here that the second term alone cannot guide MDE toward multiview consistent geometry, as it solely deals with locally fitted depths whose location and scale have already been largely altered through scale/shift fitting. Therefore, when only the second term is used, MDE has **no incentive to adapt toward canonical geometry at the global prediction level** and only adapts toward local fine details. This leads to a drop in performance, as shown in the results below. When only the second term is used for optimization (scale-shift), it performs worse in every metric than in other cases where only the first term is used (L1) or both are used in conjunction (ours). This justifies our strategy of using both losses as effective and not redundant.
>
> |  | PSNR | SSIM | LPIPS | abs_rel | sq_rel | RMSE | log_RMS |
> | --- | --- | --- | --- | --- | --- | --- | --- |
> | scale-shift | 21.43 | 0.763 | 0.330 | 0.182 | 0.109 | 0.484 | 0.205 |
> | l1 | 21.45 | 0.763 | 0.327 | 0.157 | 0.079 | 0.386 | 0.176 |
> | ours | **21.58** | **0.765** | **0.325** | **0.151** | **0.071** | **0.356** | **0.168** |
>
> > **Q4. Dataset used for ablation**
> >
>
> The quantitative results of the Ablation study in Table 3 contain **average values** of quantitative results yielded by every scene in ScanNet. It is solely for visualization purposes that we use Scene781 of ScanNet as a representative for qualitative results in Figure 8.
>
> > **Q5. Error bars**
> >
>
> We apologize that we have made a mistake in checking the checklist, as we do not show error bars in our paper. Thank you for pointing this out.
>
> [1] Uy et al., SCADE: NeRFs from Space Carving with Ambiguity-Aware Depth Estimates, CVPR 2023

---

> > ### Comment · Reviewer_Ndsp · 2023-08-18
> >
> > Thank you for your detailed responses and for addressing the questions raised. I appreciate that a distinction and clarification of the differences with SCADE is brought up in the rebuttal.
> >
> > My main concern as written in my initial review was the lack of distinction and differentiation with the existing work, SCADE. The contributions claimed in the submission did not differentiate what is novel and what is existing especially compared to the recent work. It was cited, but it did not bring up that it models ambiguity in MDE, which is the concern I raised. I do agree on their differences, and as I also pointed out in the review, the use of MDE on unseen viewpoints is novel.
> >
> > Following up on this rebuttal, as mentioned, I agree with the point on unseen viewpoints as well as computation time. I don't quite buy the argument on "orthogonality" of the two strategies. On the higher performance than SCADE, I think is less convincing since it was only shown on the scannet dataset in the original setting and the performance gap is marginal. Moreover, the ambiguities that SCADE models seem to do well on non-opaque surfaces as shown in their teaser, which is also the failure case that DaRF shows in the PDF attached in the rebuttal, which makes me think that the claim "higher performance" may not be backed that well. I do acknowledge that the official code at the time of the submission may not have been released. But the lack of comparison together with my original concern in the lack of differentiation in terms of contribution is why I gave the negative rating.

---

> > > ### Author Response · Authors · 2023-08-18
> > >
> > > Dear Reviewer Ndsp,
> > >
> > > Thanks for your reply. First of all, following your advice, we promise to strengthen our work’s comparison against SCADE in the final version of our paper, especially in regard to the MDE ambiguity removal process.
> > >
> > > However, we would need further details as to why you do not agree with our differentiation against SCADE in how they are fundamentally different and “orthogonal”: in our rebuttal, we have described our perspective on these differences in detail, such as:
> > >
> > > 1. How our work adapts (overfits) an MDE to a single specific scene to direct ambiguity removal, while SCADE increases MDE’s ambiguity for general probabilistic modeling.
> > > 2. How our work addresses & tackles the problem of MDE depth global scale fitting, in which alignment to one region of the scene inevitably leads to misalignment in many other regions. In this aspect, our additional contributions of patchwise scale-shift alignment fitting, which has proven effective.
> > > 3. Increased methodological simplicity and no need for additional MDE pretraining unlike SCADE due to our adapted MDE’s capability to directly predict absolute, canonical geometry.
> > >
> > > and we would be very grateful if you could elaborate in more detail on how you view these points are **not** in fact, differences from SCADE and thus lacking novelty. If you have any further questions regarding our methodology, we would be happy to answer them.
> > >
> > > Regarding the “marginal” performance of our model's results in comparison to SCADE that you point out, we again emphasize that our approach is orthogonal to SCADE's, achieving competitive results while using an entirely different approach that is notably simpler and more computation-efficient, as you have agreed. Moreover, we show here our additional quantitative result in comparison to SCADE using an in-the-wild dataset, and we will also add in our final paper our additional qualitative results which perform competitively to SCADE in regards to transparent surfaces you have mentioned.
> > >
> > > |  | PSNR | SSIM | LPIPS |
> > > | --- | --- | --- | --- |
> > > | DDP | 21.28 | 0.727 | 0.366 |
> > > | SCADE | 22.82 | 0.743 | **0.347** |
> > > | Ours | **22.92** | **0.760** | 0.390 |
> > >
> > > Lastly, we respectfully emphasize that our work was **concurrent** with SCADE at the time of submission. SCADE was submitted to CVPR 2023 and not yet published at NeurIPS 2023 submission nor was its code revealed. As a result, the experimental comparison was impossible at the time of writing, and its influence on our methodology was minimal. Therefore, we believe our work was under no obligation to put such a strong focus on comparison and differentiation to SCADE in our main paper, but thanks to your constructive comment we could more clearly analyze the methodological contrast between the two methods, and promise to emphasize the differences between the works in our final paper.

---

> > > > ### Comment · Reviewer_Ndsp · 2023-08-19
> > > >
> > > > Thank you for your response. and I especially appreciate the additional results.
> > > >
> > > > I agree with the two works being concurrent and that the experimental comparison is not the emphasis. However, the work was on arxiv then and DaRF did cite it. So again, the concern I had as reflected in my initial review was that even if the work was cited, the differences were not made clear (the clarification is now here in the rebuttal). By differences, it is not merely differences in terms of the approach (e.g. the advantages of DaRF over SCADE), but also in terms of contribution in "overcoming ambiguity in monocular depth estimation", where there is another work (SCADE, which you did cite) that also tackles this. How this work was cited (as shown in Ln95-96 main paper) does not bring up their modeling of ambiguities in depth -- despite being different from DaRF. I fully support the contributions and the advantages of DaRF over SCADE, but I think for the research community in general, it is important to position ones work in both current literature and clearly state contributions -- e.g. "new" vs "better approach". The clarifications and additional results in the rebuttal do provide support for the latter.
> > > >
> > > > This relates to the point of "orthogonality", I think our disagreement might be on the definition of how two problems are orthogonal. From my perspective, two approaches are considered orthogonal when they are unrelated/independent of each other, e.g. if you have one approach that improves depth estimation and another that improves normals. Then typically, the two approaches can be used together to complement each other. So an argument on orthogonality usually comes with a showcase or proposed experiment that two approaches can be complementary to each other. I think the points that you brought up are **differences** between the MDE of DaRF and SCADE, which I totally do agree with -- I was not questioning that the two approaches are the same.

---

> > > > > ### Author Response · Authors · 2023-08-20
> > > > >
> > > > > Dear Reviewer Ndsp,
> > > > >
> > > > > Thank you for your clarification. We completely agree with your statement that in our main paper, the discussion of comparisons & differences between our work and SCADE in regards to tackling “overcoming ambiguity in monocular depth estimation” was limited. We promise to improve this in the final manuscript. Due to your constructive comments, we could analyze in further detail the differences between the two methods, and these comparisons will be thoroughly included in the final version of our paper. We will position our work more clearly as a “better approach” in our paper, according to your advice, and include the additional comparison results. Such in-depth discussions are both necessary and will greatly strengthen our paper.
> > > > >
> > > > > Regarding the term “orthogonality”, we also believe our disagreement stems from the difference in the definition we had of the term - in our rebuttal, as you have said, we were using this term as similar to **differences**. When it comes to the definition you give - how the two approaches can be complementary to each other - our usage of this term was not in this manner, and we agree with your statement that our work should be classified as a “better approach”. We will also take this discussion into account when revising the final version of our paper.
> > > > >
> > > > > Lastly, in your final decision, we would be truly grateful if you could take into account our work’s different contributions and advantages in comparisons SCADE that we have clarified, as well as the additional quantitative results, which we will thoroughly incorporate in our final manuscript. We thank you for the detailed and careful review you have given us.

---

> > > > > > ### Comment · Reviewer_Ndsp · 2023-08-20
> > > > > > **Response to the Authors**
> > > > > >
> > > > > > Thank you for your response. I appreciate that proper discussion and differentiation (both in method contribution and in handling/modeling of MDE ambiguity) of SCADE will be made in the final version. I acknowledge the contributions of DaRF, and I am happy to raise my rating.

---

> > > > > > > ### Author Response · Authors · 2023-08-21
> > > > > > >
> > > > > > > Dear Reviewer Ndsp,
> > > > > > >
> > > > > > > Thank you very much for your recognition and for increasing your score. We are happy to include the discussion of comparisons & differences between our work and SCADE in the final version of paper.
> > > > > > >
> > > > > > > Regards, Authors

---

### Official Review · Reviewer_KpuH · 2023-07-06

**Soundness:** 3 good
**Presentation:** 3 good
**Contribution:** 3 good
**Rating:** 6
**Confidence:** 5

**Summary:**

This paper addresses the problem of few-shot NeRF reconstruction. The authors propose using monocular depth estimation (MDE) networks to provide geometry prior to NeRF at both seen and unseen viewpoints. They propose overcoming the ambiguity problems associated with monocular depths by MDE adaption. Experimental results deomonstrate improvements in both rendered novel views and rendered depths.

**Strengths:**

+ Unlike previous works which only exploit depth information at seen viewpoints, this work also exploits MDE to constraint NeRF at unseen viewpoints, leading to more robust and coherent NeRF optimization. The authors demonstrate through an example that the strong geometric prior within the MDE model enables it to generate reliable depth from noisy NeRF rendering. This makes MDE at unseen viewpoints feasible.
+ The proposed patch-wise scale-shift fitting helps reducing the impact of erroneous depth differences generated by MDE networks in distilling the monoclular depth prior to the NeRF.
+ Adapting MDE to absolute scene geometry embedded in NeRF further helps to resolve ambiguities in surface curvature and orientation, and improve multiview consistency.
+ The proposed method demonstrates sota results, both qualitatively and quantitatively, on two real-data sets, particularly showing superb results in rendered depths in few-shot NeRF.
+ Ablation study has been included to demonstrate the effectiveness of each major deisgn component.
+ Overall this paper is well-written and well-organized. It is easy to follow.


**Weaknesses:**

- In MDE adaption, it is not clear why the monocular depths predicted from the rendered input views instead of that predicted from the input views are adopted in (6). There is no explanations or disucssions on the effect of choosing between these two. There is also no explanations or discussions on why only input views are considered.
- In confidence modeling, why the predicted depth of a point in an input view (after scaling ana shifting) is directly compared with its rendered depth in a target view in (9)? Note both predicted depth and rendered depth are measured with respect to the viewpoints.

**Questions:**

- Figure 2 is rather confusing. It is not clear how one should interpret / understand the renderings. More descriptive caption should be provided.
- Are there any failure cases? Any analysis for the causes of failure?
- What is the minimum number of viewpoints for the proposed model to produce reasonable reconstruction?

**Limitations:**

Limitations and impact are only included in the supplementary material, but not in the main paper.

---

> ### Author Rebuttal · Authors · 2023-08-10
>
> > **Q1. Details of MDE adaptation**
> >
>
> Thank you for pointing this out. Equation 6 has a **notation mistake** on our part, as we do use monocular depths predicted from ground truth input image, $D^*_{i}$, and not the one from depth rendered from NeRF, $\bar{D}^*_{i}$. Our accurate methodology, in which we use the depth predicted with MDE from ground truth input images, is correctly described in Figure 1. We apologize for this mistake and we promise to revise it in the final version of our paper as the following:
>
> $
> \mathcal{L}_\text{MDE} = \sum\_{I\_i\in\mathcal{S}} \sum\_{\textbf{p} \in \mathcal{P}} \left(||\texttt{sg} \left(\bar{D}\_i(\textbf{p})\right) - D^*\_i(\textbf{p})|| +  ||(w\_i\texttt{sg}\left(\bar{D}\_{i}(\textbf{p})\right)+q\_i) - D^*\_i(\textbf{p}) || \right)
> $
>
> We used only input viewpoints for MDE adaptation, since using rendered color patch in unseen viewpoint as supervision might lead MDE to lose its geometric prior due to its noisy rendering results.
>
> > **Q2. Details of Confidence Modeling**
> >
>
> Thank you for pointing this out. Your comment is correct, and upon closer inspection, we have made a mistake in our thresholding equation so it does not correctly describe our masking process. We apologize for our mistake, and we are also very thankful for your careful reading and revision of our paper. The correct thresholding equation is as follows:
>
> $
> M\_i(\textbf{p}) = \big[\|R\_{i}(w\_i{D}^*\_i(\textbf{p}) + q\_i)K^{-1}\textbf{p} - R\_{l}\bar{D}\_l\textbf{p}')K^{-1}\textbf{p}' \| < \tau\big]
> $
>
> Where $R_{i}$ and  $R_{l}$ stand for camera-to-world extrinsic matrices that transform the 3D points from camera view spaces to the canonical space. In short, using the predicted depth of viewpoint $i$ and NeRF-rendered depth of target viewpoint $l$, we compare the 3D point acquired from both viewpoints to canonical space by calculating the distance between the two. Only if the distance is lower than the threshold (in agreement with each other), do we take the mask as reliable. Again, thank you for your constructive revision, and we promise to revise this mistake in the final version of our paper.
>
> > **Q3.  Details of Figure2**
> >
>
> Thank you for your constructive comment. We agree that Figure 2 is difficult to recognize at first sight, and we promise to fix it.
>
> Our intention for Figure 2 was to visualize our point that a single scale-shift value cannot perfectly fit into ground truth geometry due to the ambiguity of distances between object instances. For this reason, we have visualized **error** of a predicted point cloud of a room from the bird’s eye view to show that patch-wise fitting most accurately fits point cloud to ground truth geometry. This is done with jet color coding, so that red color means large error and blue color means small error. This point cloud is projected depth from the input image from the viewpoint stated as red camera.
>
> Following your comment, we promise to either replace it with Figure 1 of our supplementary material (which describes the same phenomenon more clearly) or reinforce it with an additional caption that explains it in a more detailed manner.
>
> > **Q4.  Any Failure cases?**
> >
>
> For failure cases, there are occasions when our confidence modeling fails and unable to completely filter out erroneous predictions. Also, in out-of-domain cases where MDE fundamentally models depth incorrectly, our model also shows drop in performance as well. Failure case is shown in Figure2 of the attached pdf file, showing the case of wrong depth prediction in the window since neither NeRF and MDE model predict accurate depth of the transparent object.
>
> > **Q5. Minimum number of viewpoints?**
> >
>
> We do not analytically calculate the minimum number of viewpoints required for few-shot reconstruction, as that number will vary scene by scene according to each scene’s shape, geometry and occlusions. Instead, like all other few-shot NeRF methods [1,2,3,4], we assume a setting supposed by all other few-shot NeRF methods - an extreme scenario where the known viewpoints hardly overlap with each other yet all directions of the scene are viewed at least once - which is 10 viewpoints in our case. If we reduce the number of viewpoints from here, scene reconstruction will still happen, but it will result unseen directions / regions of the scene where no information is provided whatsoever. In such cases our model cannot perform reconstruction, as it is not a generative model capable of imagining unseen parts and our model is not designed to do so in the first place. Furthermore, visualization of camera setting for few-shot is visualized at Figure 1 of the attached pdf file.
>
> [1] Jain et al., Putting NeRF on a Diet: Semantically Consistent Few-Shot View Synthesis, ICCV 2021.
>
> [2] Kim et al., InfoNeRF: Ray Entropy Minimization for Few-Shot Neural Volume Rendering, CVPR 2022.
>
> [3] Niemeyer et al., RegNeRF: Regularizing Neural Radiance Fields for View Synthesis from Sparse Inputs, CVPR 2022.
>
> [4] Yang et al., FreeNeRF: Improving Few-shot Neural Rendering with Free Frequency Regularization, CVPR 2023.

---

> > ### Comment · Reviewer_KpuH · 2023-08-15
> >
> > The authors have addressed all my concerns in their rebuttal. After taking the comments of fellow reviewers into account, I would like to keep my original recommendation of "weak accept".

---

### Official Review · Reviewer_oMo1 · 2023-07-06

**Soundness:** 3 good
**Presentation:** 3 good
**Contribution:** 3 good
**Rating:** 5
**Confidence:** 3

**Summary:**

The paper presents a new few-shot neural radience field approach based on joint monocular depth adaption. The main idea of the proposed approach is to utilize the monocular depth estimator to improve the geometry prior of NeRF representation. The motivation is reasonable. Also, it presents attractive performance on both indoor and outdoor scenes.

**Strengths:**

1. The idea of utilizing monocular depth estimator is interesing and the depth estimator can provide reasonable geometry information for the scene reference.

2. The proposed approach has provided attractive performance on the real-world benchmarks. Sufficient ablations have been conducted to validate the design of the proposed algorithm.

3. The presentation of the paper is good.

**Weaknesses:**

1. The paper is based on SCADE[49]. According the results in Table 1, the proposed approach is lower than [49] on the LPIPS score. Is there any potential explanation of this result? Also, How about the results on the "in-the-wild benchmark" proposed in [49]?

2. How is case when the testing data is out of the distribution of the depth estimator? For example, in the case when the depth estimator failed to predict the accurate depth information.

3. How about the inference speed of the proposed algorithm?

**Questions:**

Please address the comments released in the weakness section.

**Limitations:**

The authors have not full discussed the limitations of the proposed approach.

---

> ### Author Rebuttal · Authors · 2023-08-10
>
> > **Q1. Comparison with SCADE**
> >
>
> Thank you for your comment. First of all, we would like to emphasize that our paper is **not based** on SCADE [1] nor use it as its baseline but instead suggests a method orthogonal to it. There is a fundamental difference between our work and SCADE’s approach toward handling ambiguity present within MDE: our work proposes an online, scene-specific adaptation of MDE which **directly adapts it to predict canonical geometry** in accordance with NeRF under optimization. On the other hand, SCADE **injects uncertainty into MDE** through additional pretraining so that canonical geometry can be estimated through probabilistic modeling between multiple modes of estimated depths. So, while the ultimate goal regarding our usage of MDEs may be similar (ambiguity removal), the core idea behind them can be seen as more or less opposite.
>
> In this light, we explain our potential explanation for lower LPIPS score in Table 1: our work’s baseline, K-planes [2], has been shown to perform slightly weaker in LPIPS than **NeRF-pytorch** [3], which is the baseline for SCADE. We have chosen K-planes as our baseline due to its fast optimization speed and more efficient memory, but due to this trait of baseline, we believe that our model show slightly less performance in LPIPS metric than SCADE, while exceeding it in all other metrics. At the time of our submission, the code for SCADE was not revealed, so we could not directly perform experiments with SCADE or its baseline: now it has been revealed, we bring you quantitative experiment results for accurate comparison.
>
> Here are the results of our model, DäRF, for in-the-wild benchmark proposed in SCADE, which shows that our model is slightly better results than SCADE in two metrics.
>
> |  | PSNR | SSIM | LPIPS |
> | --- | --- | --- | --- |
> | DDP | 21.28 | 0.727 | 0.366 |
> | SCADE | 22.82 | 0.743 | **0.347** |
> | Ours | **22.92** | **0.760** | 0.390 |
>
> > **Q2. How to deal with errors in MDE**
> >
>
> Due to MDE’s weakness in predicting geometry for data outside of distribution, this method does show a drop in performance when faced with testing data out of the distribution of depth estimator. However, we also point out that this is a fundamental weakness in all NeRF methods that utilize MDE, and the strength of our method derives from the fact that our method allows the adaptation of off-the-shelf pre-trained MDE to a scene without any additional training or dataset, unlike SCADE.
>
> > **Q3.  Inference speed of our model**
> >
>
> The inference speed of our model is identical to that of K-planes, which is our baseline model.
>
> [1] Uy et al., SCADE: NeRFs from Space Carving with Ambiguity-Aware Depth Estimates, CVPR 2023
>
> [2] Fridovich-Keil et al., K-Planes: Explicit Radiance Fields in Space, Time, and Appearance, CVPR 2023.
>
> [3] Lin Yen-Chen. Nerf-pytorch. 2020.

---

### Official Review · Reviewer_Eioc · 2023-07-10

**Soundness:** 2 fair
**Presentation:** 3 good
**Contribution:** 2 fair
**Rating:** 5
**Confidence:** 3

**Summary:**

The paper proposes a sparse-view NeRF framework that jointly trains a NeRF with a monocular depth estimator. By adapting the MDE network to the target scene, the predicted depth will provide better geometry prior for the NeRF model.

**Strengths:**

The joint training of MDE and NeRF improves the model's ability compared to existing methods.
After training both models, we'll have a better NeRF that renders more accurate novel views and also a more accurate mono depth estimator.

**Weaknesses:**

- For the few-shot NeRF setting, since we hope to train the NeRF directly, we shall have access to camera poses for multi-views, right?
If that's the case, what's the benefit of using monocular depth estimators, compared to multiview stereo networks such as MVSNet?
- If we use COLMAP to obtain the camera poses, we should also be able to construct a point cloud. What's the benefit of using the proposed distillation mechanism, compared to overcoming the scale and shift ambiguity problem of mono depth with depths extracted from point clouds?
- How is Tab. 2 calculated? Are they evaluated on the trained views of ScanNet?

**Questions:**

Please refer to weaknesses.

**Limitations:**

Please refer to weaknesses.

---

> ### Author Rebuttal · Authors · 2023-08-10
>
>
> > **Q1. Benefits of MDE compared to MVS networks**
> >
>
> Thank you for pointing this out. It is true that MVS networks can be used for better NeRF training, as shown in previous works [1, 2]. However, their approach of leveraging image features combined from multiple viewpoints serves as a critical weakness in more extreme wide-baseline scenarios where the number of input viewpoints is very limited and the distances between known cameras are large, due to increased difficulty in finding consensus between image features - resulting in a sharp drop in performance.
>
> Our methodology of leveraging monocular depth networks’ powerful geometric prior helps overcome such weaknesses of MVS methods. As MDE predicts geometric information for each image independently, which fundamentally bypasses wide-baseline scenario problems of previous MVS methods. In addition, our MDE regularization method can be **directly extended to unseen viewpoints** for stronger regularization and artifact removal, which is impossible for standard MVS methods that only operate upon input viewpoints.
>
> In this way, our paper demonstrates how these desirable qualities of MDE can be effectively leveraged in sparse-view NeRF settings in a complementary manner, orthogonal to previous MVS methods and effectively improving upon their weaknesses.
>
> > **Q2. Comparison to using COLMAP point cloud for scale-shift fitting**
> >
>
> It is true that the point cloud acquired through COLMAP provides us with absolute depth information, and is beneficial for aligning the scale-shift value of the predicted depth, especially in the acceleration of the early stages in training.
>
> However, since COLMAP point cloud only provides sparse depth information, it falls short in modeling local (as described in our paper, patch-wise) scale-shift values, as the points are unequally distributed in the point cloud and the sparse regions do not provide our model enough geometric information for patch-wise scale-shift value modeling. Since local MDE alignment is precise solution than global scale-shift fitting [3], we find that solely using dense information predicted by MDE allows us to robustly capture patch-wise scale-shift values for depth fitting regardless of where the patch is sampled.
>
> Also, because COLMAP point clouds are generated from input images, they are severely limited from providing our model with accurate geometric data in unseen regions, which makes it unfit for out unseen viewpoint depth regularization loss. Therefore, we find that using rendered depth data captured by NeRF provides us stronger advantages than naïvely employing COLMAP point cloud to overcome the scale and shift ambiguity problem of monocular depths.
>
> > **Q3. Details of evaluation method in Tab.2**
> >
>
> We evaluated adapted MDE model only on test views of ScanNet. Table 2 is for evaluating view consistency of MDE model [4], so we utilize a single scaling factor s for each scene, which is the median scaling value averaged across all test views.
>
> [1] Wei et al., NerfingMVS: Guided Optimization of Neural Radiance Fields for Indoor Multi-view Stereo, ICCV 2021
>
> [2] Chibane et al., Stereo Radiance Fields (SRF): Learning View Synthesis for Sparse Views of Novel Scenes, CVPR 2021
>
> [3] Zhang et al., Hierarchical normalization for robust monocular depth estimation, NeurIPS 2022
>
> [4] Zhang et al., Consistent depth of moving objects in video, SIGGRAPH 2021.

---

### Author Rebuttal · Authors · 2023-08-10

# General Response

We would like to first thank all the reviewers for their helpful suggestions and constructive reviews. We are greatly encouraged by their assessment of our work as **well-motivated** (VhkU), with novel and **interesting** (oMo1, VhkU) findings of effectively **resolving MDE’s ambiguities** (KpuH, VhkU), and its exploitation at **unseen viewpoints** (Ndsp, KpuH, VkhU) thus **improving upon previous models** (Eioc, KpuH). The reviews assess our work as displaying **state-of-the-art** (KpuH) and **attractive performance** (oMo1) on **real-world** (KpuH, oMo1) benchmarks, well-supported by **sufficient ablation** (oMo1, KpuH, Ndsp) and **good presentation** (oMo1, KpuH). We are grateful that they saw significance in our quantitative and qualitative improvement over our baselines, achieving few-shot novel view synthesis quality competitive to current SOTA models. We carefully address each concern given by reviewers with detailed explanations and supporting experimental results.

We clarify our work’s distinction from SCADE [1], which has been brought up by multiple reviewers due to its similar motivation of exploiting MDE for few-shot NeRF. Our work proposes an online, scene-specific adaptation of MDE which **directly adapts it to predict canonical geometry** in accordance with NeRF under optimization. On the other hand, SCADE **injects uncertainty into MDE** so that canonical geometry can be estimated through probabilistic modeling between multiple estimated depths. Therefore, while the two works’ ultimate goal regarding MDEs is similar (ambiguity removal), the core ideas behind them are orthogonal. Another important distinction is that our method of MDE adaption allows it to be leveraged to **predict depths of unknown viewpoints for artifact removal and geometry regularization**, which is a novel contribution of our method. We further elaborate on this distinction with much deeper detail in our first response to Reviewer Ndsp.

[1] Uy et al., SCADE: NeRFs from Space Carving with Ambiguity-Aware Depth Estimates, CVPR 2023

---

### Comment · Area_Chair_ALqT · 2023-08-12
**Author-reviewer discussion starts**

Dear reviewers,

Thanks for serving as reviewers.

The authors have submitted a rebuttal including a PDF with tables and figures. Please review through the rebuttal and reviews from other authors. If you have any questions, please feel free to let the authors know. You are more than welcome to post comments for further explanation or clarification before 1pm EDT on 8.21.

Best,
AC

---

### Comment · Area_Chair_ALqT · 2023-08-18
**Kind reminders for reviewers**

Dear reviewers,

If you have not responded to the authors' feedback, please take some time to read through their responses and reviews from other reviewers. We would be very pleased to hear your thoughts.

Thanks,

AC

---

### Decision · Program_Chairs · 2023-09-21

**Decision:**

Accept (poster)

**Comment:**

This submission received 5 positive recommendations. Initially, the reviewers were concerned about the limited contribution, the difference to SCADE, and the errors in MDE. The authors addressed most of the concerns in the rebuttal and the responses to reviewers’ additional comments. The reviewers reached a consensus of acceptance after the discussion period. The AC read through the submission, the review, the rebuttal, the confidential comments to the AC and the discussions. The AC agrees with the reviewers that the idea is interesting and novel and the evaluation is convincing. Per this, the AC supports the reviewers’ decision, and accepts this submission. The decision was discussed with and approved by the SAC. Please follow the reviewers’ comments (especially, pay more attention to reviewer Ndsp’s comments and advice) to improve the manuscript.